# Highly parallel and efficient single cell mRNA sequencing with paired picoliter chambers

Mingxia Zhang [1,7], Yuan Zou[1,2,7], Xing Xu[1,7], Xuebing Zhang[3], Mingxuan Gao[1], Jia Song[4], Peifeng Huang[1], Qin Chen[3], Zhi Zhu[1], Wei Lin[5,6], Richard N. Zare [2] & Chaoyong Yang [1,4✉]

ScRNA-seq has the ability to reveal accurate and precise cell types and states. Existing scRNA-seq platforms utilize bead-based technologies uniquely barcoding individual cells, facing practical challenges for precious samples with limited cell number. Here, we present a scRNA-seq platform, named Paired-seq, with high cells/beads utilization efficiency, cell-free RNAs removal capability, high gene detection ability and low cost. We utilize the differential flow resistance principle to achieve single cell/barcoded bead pairing with high cell utilization efficiency (95%). The integration of valves and pumps enables the complete removal of cell-free RNAs, efficient cell lysis and mRNA capture, achieving highest mRNA detection accuracy (R = 0.955) and comparable sensitivity. Lower reaction volume and higher mRNA capture and barcoding efficiency significantly reduce the cost of reagents and sequencing. The single-cell expression profile of mES and drug treated cells reveal cell heterogeneity, demonstrating the enormous potential of Paired-seq for cell biology, developmental biology and precision medicine.

[1] State Key Laboratory of Physical Chemistry of Solid Surfaces, The MOE Key Laboratory of Spectrochemical Analysis & Instrumentation, Key Laboratory for Chemical Biology of Fujian Province, Collaborative Innovation Center of Chemistry for Energy Materials, Department of Chemical Biology, College of Chemistry and Chemical Engineering, Xiamen University, Xiamen 361005, P. R. China. [2] Department of Chemistry, Stanford University, Stanford, CA 94305, USA. [3] Hangzhou Weizhu Biological Technology Co., Ltd, Hangzhou, China. [4] Institute of Molecular Medicine, State Key Laboratory of Oncogenes and Related Genes, Renji Hospital, School of Medicine, Shanghai Jiao Tong University, Shanghai, China. [5] Translational Genomics Research Institute, Molecular Medicine Division, Phoenix, AZ, USA. [6] Hunan Provincial Key Lab of Emergency and Critical Care, Hunan People's Hospital, Changsha, China. [7] These authors contributed equally: Mingxia Zhang, Yuan Zou, Xing Xu. ✉email: cyyang@xmu.edu.cn

Many physiological functions of multicellular organisms are reflected in the temporal and spatial changes in gene expression between constituent cells[1]. Cellular heterogeneity presented by different gene expression profiles, functions and morphologies occurs not only in different tissues but also even within the same cell type. Transcriptomic profiling of individual cells has emerged as an essential tool for characterizing cellular diversities to have a complete catalog of cell types or their functions. However, traditional single-cell analysis methods can monitor only a few types of molecules for each cell[2,3]. In 2009, single cell mRNA sequencing (scRNA-seq) was first introduced by Tang to analyze the whole transcriptome in single cells[4]. As one of the most powerful tools to understand the heterogeneity of biology[5–10], scRNA-seq contributes to discovering the cellular and molecular driving forces of biology, unveiling new biological insights about cell types[11–16], which has a broad impact on diverse biology fields, including development[17,18], immunology[19,20], neurobiology[20], cancer[8,21–23], gene regulation[24], and epigenetics[6,25].

For scRNA-seq, it comes first with the isolation of single cells from their native environment, such as a culture dish or cell suspension. Traditional methods, including limiting dilution[26], capillary picking[27], and laser capture microdissection (LCM)[28], suffer from time and labor consumption, cell damage, and low throughput. In recent years, microfluidic devices characterized by their manipulation integration, low reagent consumption, size/volume compatibility, and external contamination isolation have demonstrated their capability in high-efficiency, high-viability, and low-cost single-cell isolation. After single-cell isolation, each cell must be processed and sequenced individually to obtain transcriptome information, which is labor intensive and cost prohibitive, especially when a large population of cells is needed to be processed. To address this problem, several novel high-throughput platforms have been reported, including Drop-seq[13], inDrop[12], Seq-well[15], and Microwell-seq[14], etc., which used barcoded beads to label individual cells during reverse transcription so that cDNAs from all the cells could be simultaneously pooled for amplification and sequencing[29]. By identifying the cell barcode and molecular index, the cell origin of cDNA could be inferred and the amplification bias could be corrected.

Successful barcoding of individual cells relies on co-encapsulation of a single cell and barcoded bead within a single droplet or microwell[30]. Current high throughput scRNA-seq platforms utilize a limited dilution strategy for cell/bead encapsulation to ensure that there is no more than one cell or one bead in each reaction compartment based on Poisson statistics. Unfortunately, such a limiting dilution strategy for both cell and bead is wasteful of reagents and causes loss of cells, which is unacceptable when only a limited number of cells, such as stem cells, neuron cells, or circulating tumor cells (CTCs) are available[10]. Additionally, how to avoid the interference of cell-free RNAs produced during the preparation of cell suspension to achieve information about the true original cell is another challenge for scRNA-seq[13]. For the preparation of solid tissues, enzymatic digestion will destroy the extracellular matrix and disrupt the cell–cell junctions, releasing RNAs into the extracellular "soup". Furthermore, cell death also results in the release of cellular RNAs in both tissues and blood samples. These cell-free RNAs would lead to noise in the data produced by scRNA-seq experiments. Thus, it is difficult to evaluate how faithfully the tissue and blood samples are represented by the scRNA-seq analysis.

In order to realize both a parallel and an efficient processing for a limited number of cells, it is very necessary to develop a high-efficient single cell manipulation platform for scRNA-seq. Herein, we present a scRNA-seq platform, named Paired-seq, with high cells/beads utilization efficiency[31], as well as excellent sequencing accuracy and sensitivity by integrating barcoding technology for cell tagging, droplet strategy for parallel compartmentalization, hydrodynamic differential flow resistance based isolation for single cell/bead, and micro-pumping structure for active fluidic control. Our Paired-seq chip allows automatic isolation enabling the pairing of single cell and single bead in a reaction unit with an efficiency up to 95%. Thus, Paired-seq achieves efficient utilization of precious cells. After cell/bead capture and pairing, formation of picoliter droplets allows highly parallel processing of dozens to thousands of cells. Integration of valves and pumps enables the on-chip removal of cell-free RNAs in the cell chambers, making it possible to identify the true composition of the original sample. Cell lysis, mRNA capture and reversed transcription can be efficiently carried out in the tiny droplet by virtue of active pumping for fluid transportation and rapid mixing. Analysis results of sequencing data for External RNA Controls Consortium (ERCC) suggests that our method offers high accuracy ($R = 0.955$) and comparable sensitivity compared to other current scRNA-seq platforms. What is more, the lower reaction volume and higher mRNA capture and barcoding efficiency significantly reduce the cost of reagent and the sequencing cost. Using Paired-seq, we analyze the single-cell expression profile of mES cells and anti-cancer drug treated cells, revealing the heterogeneity of the cell population during differentiation and drug treatment processes which show an enormous potential of our platform for cell biology, developmental biology and precision medicine.

## Results

**Workflow of Paired-seq**. We designed and fabricated a microfluidic chip (Fig. 1a and Supplementary Figs. 1–3) which contained hundreds to thousands of reaction units (Fig. 1c and Supplementary Fig. 2) for parallel single cell and single barcoded bead pairing and sample processing (Supplementary Movie 1). Each reaction unit is designed based on the hydrodynamic differential flow resistance principle to allow no more than one bead and cell to be captured in each bead capture chamber and cell capture chamber, respectively (Fig. 1d, a). The cell-free RNAs can be easily removed by injecting washing buffer while maintaining the single cells in the chambers. After bead and cell isolation, gas is introduced to form two droplets in each reaction unit containing one cell and one bead, respectively (Fig. 1d, b). These two droplets are then merged by turning off a separation valve located in between, thus forming a larger picoliter droplet containing exactly one bead and one cell (Fig. 1d, c). Because the bead-loading solution contains cell lysis buffer, mixing of the bead droplet with the cell droplet leads to cell lysis. The barcoded bead contains cell label and molecular index for cell/molecular barcoding, poly-$(dT)_{30}$ for mRNA capture and universal primer for cDNA amplification (Fig. 1b). Once released from the cell, poly A-tail mRNAs are captured by poly-$(dT)_{30}$ on the beads and reverse-transcribed to form the cDNAs (Fig. 1e). After subsequent bead recovery, cDNA amplification, library preparation and sequencing, the original information about the cell/molecule can be inferred to achieve an expression matrix for dozens to thousands of single cells (Fig. 1f, g).

**Chip design for Paired-seq**. The three-layer chip consists of a capture channel layer, a valve/pump actuation control layer, and an elastomeric membrane layer in between (Fig. 2a and Supplementary Figs. 1–3). Each chip contains hundreds of reaction units consisting of a cell flow channel and bead flow channel connected by a connection channel (Fig. 2b). To break the limitation of limiting dilution and avoid wasting precious cells, Paired-seq chip

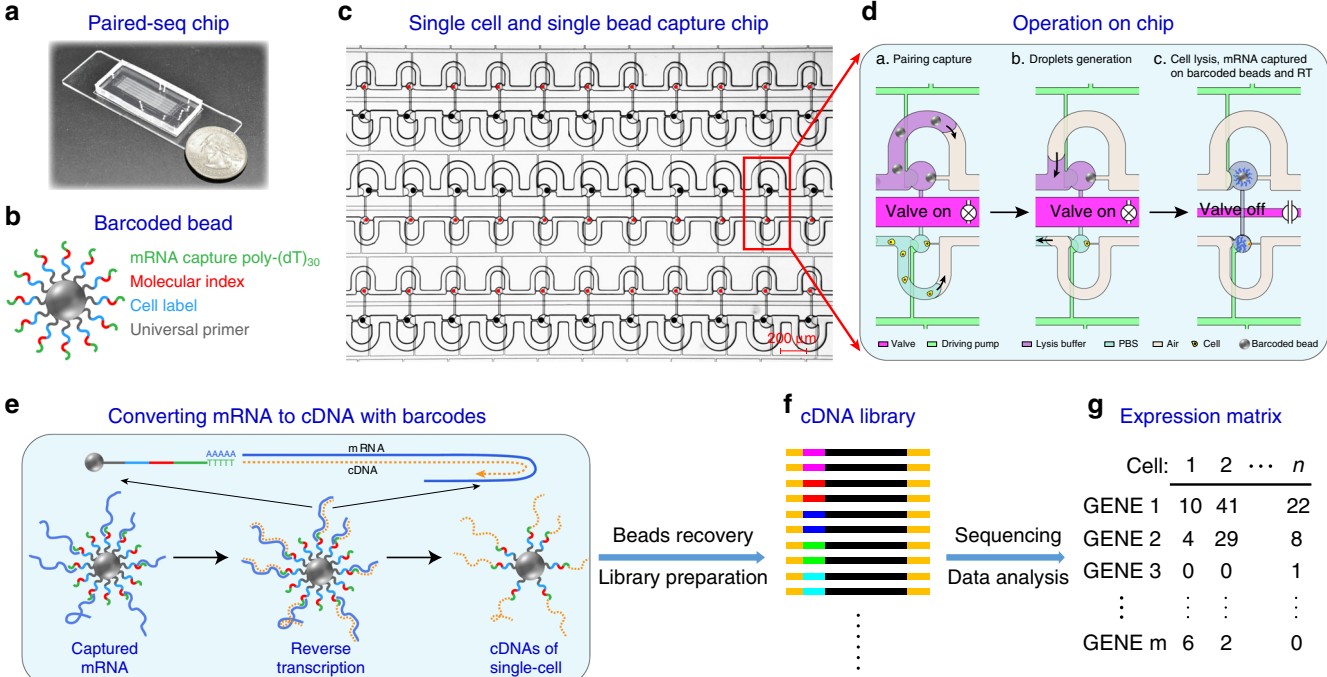

**Fig. 1 Paired-seq: a Platform for DNA Barcoding scRNA-Seq. a** Photograph of Paired-seq chip with a Quarter dollar coin. **b** Sequence of primers on the barcoded beads. The primers on beads contain mRNA capture poly-(dT)$_{30}$, molecule index, cell label, and universal primer. **c** Schematic diagram of cell and barcoded bead pairing on Paired-seq chip. Scale bar is 200 μm. **d** Schematic of the basic workflow for single cell and bead parallel manipulation on Paired-seq chip, including (**a**) capture and pairing of single cells and single beads (**b**) droplets generation to separate adjacent units (**c**) cell lysis and mRNA captured on barcoded beads. **e** After hybridizing to the primers on the barcoded beads, mRNAs are reverse-transcribed to produce cDNAs. All the cDNAs-attached beads are recovered from the chip and (**f**) subsequently amplified for library preparation in bulk. **g** Data analysis to generate single cell expression matrix. Millions of paired-end reads are generated from a Paired-seq library on a high-throughput sequencer. The reads are first aligned to a reference genome to identify the gene of origin of the cDNA. Next, reads are grouped by their cell barcodes, and individual UMIs are counted for each gene in each cell. The result is a "digital expression matrix" that each column corresponds to a cell, each row corresponds to a gene, and each entry is the integer number of transcripts detected from that gene, in that cell.

is designed based on a paired differential flow resistance capture principle, so that a single cell

and a single bead can be isolated and paired with high efficiency. In the sample loading process, when a capture chamber is empty, flow resistance along the straight channel is lower than that in the long loop bypass channel, and the main stream flows along the straight channel, leading to a single cell/bead in the flow being trapped in the chamber (Fig. 2b, Trapping mode). The size of the trapped cell/bead is larger than that of the orifice of the capture chamber and thus will block the local flow and then dramatically increase the flow resistance along the straight channel. Consequently, the main flow redirects to the bypassing channel and subsequent cells/beads will flow into the bypassing stream, going to the next paired unit (Fig. 2c, Bypassing mode). This capture mechanism ensures that there is no more than one cell/bead captured in one chamber. Because the diameter of beads (20–40 μm) is larger than that of cells, an asymmetrical paired unit is designed with a wider channel for beads and a narrower channel for cells.

To allow efficient cell lysis and mRNA capture to afford high mRNA detection sensitivity, valves and pumps were integrated in the Paired-seq chip. Firstly, to enable independent loading of cell and bead solutions, a blocking valve is designed orthogonally below the connection channel for each paired unit (Supplementary Movie 2). Secondly, as each chip consists of hundreds to thousands of reaction units, to avoid cross-contamination, the reaction unit is separated by air. This can be realized by reversely introducing an air flow in the capture channel and extra solution outside capture chamber is dispelled, forming water-in-air

droplets and effectively separating each individual reaction unit (Fig. 2d, Droplets forming mode and Supplementary Movie 3). Formation of droplets allows hundreds of cells to be processed in parallel for high-throughput analysis. Finally, to facilitate the exchange of reagents between the paired chambers droplets, there is a driving pump below each capture chamber (Fig. 2a, e). By alternately activating the driving pumps for the cells and the beads, solutions in the two chambers can be easily transferred back and forth, thus allowing efficient mixing of the paired droplets.

To better understand the flow characteristics around the microfluidic traps and to determine the optimal parameters for microfluidic channel design, a computational fluid dynamics (CFD) analysis was carried out using COMSOL 4.3 (COMSOL Multiphysics) to simulate the hydraulic resistance in the channel of the paired unit in the mode of trapping, bypassing and droplets forming, respectively (Fig. 2b–d).

**High efficiency of single-cell assays on Paired-seq chip.** In order to demonstrate the feasibility of the chip design for scRNA-Seq, a Paired-seq chip with 800 and 2000 units was first fabricated (Fig. 2f–h and Supplementary Fig. 2). Based on the dimensions of the fabricated chip, the total volume of each reaction unit was calculated to be less than 400 picoliter.

To test the single cell/bead isolation and pairing efficiency of the Paired-seq chip, barcoded beads and Calcein AM-strained K562 cells were loaded. Supplementary Movie 4 and Movie 5 illustrate the dynamic process of single-bead and single-cell

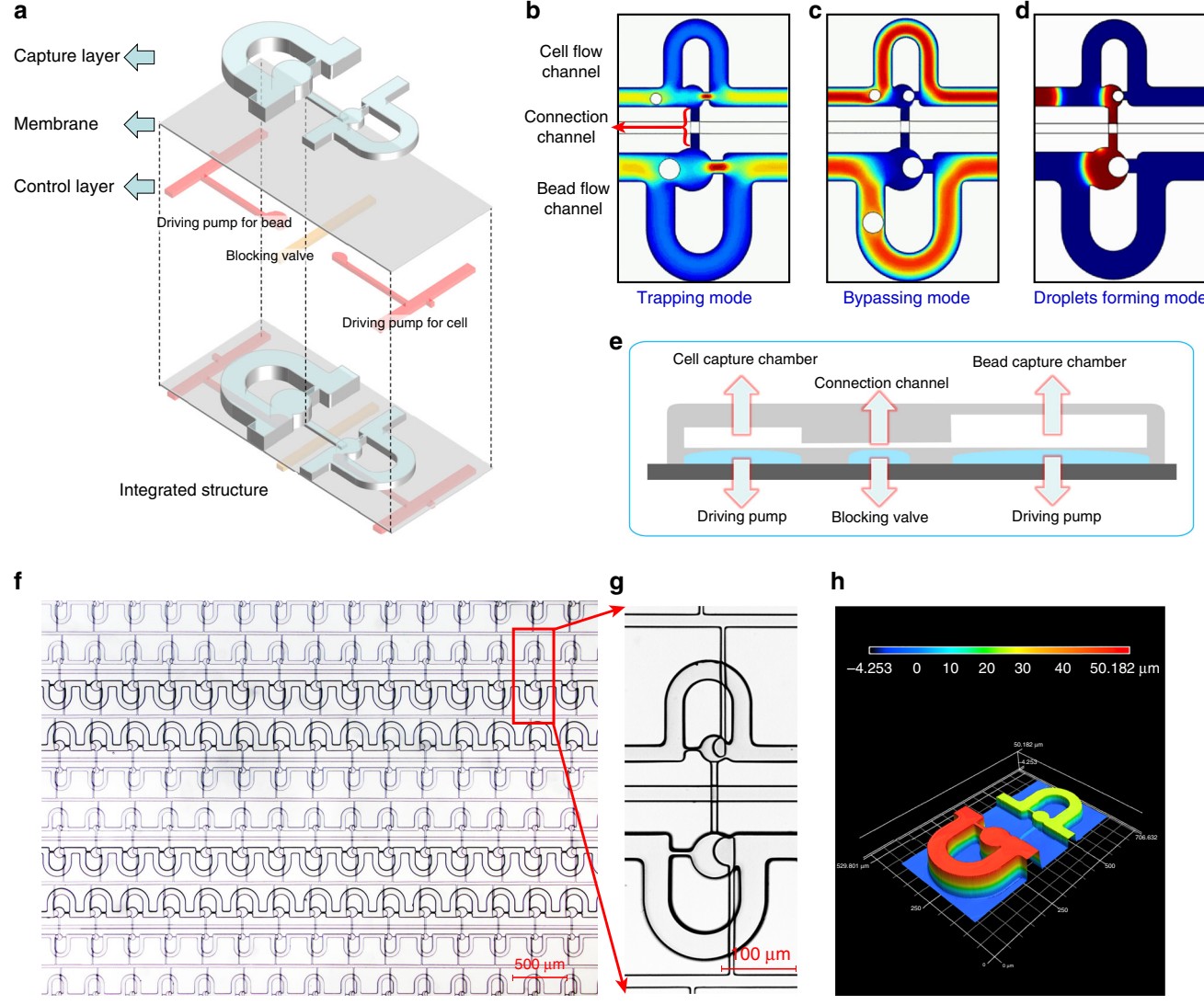

**Fig. 2 Design criterion and structure characterization of Paired-seq chip. a** The 3D cartoon diagram of the capture layer and the control layer of the chip. **b, d** The simulation results of trapping mode (**b**), bypassing mode (**c**) and droplets forming mode (**d**). **e** The cartoon diagram of cross section for one unit in Paired-seq chip. **f, g** Structure characterization of Paired-seq chip. Top view image of a Paired-seq chip with 800 units (**f**) and one paired unit (**g**). **h** The 3D surface profiling of SU8 silicon model of the flow layer for one paired unit by 3D laser scanning confocal microscopy.

trapping, confirming that the trapped cells/beads work as plugs to block the local flow and prevent the incoming of subsequent cells/beads. As expected, successful single cell/bead trapping and pairing was observed with high efficiency (Fig. 3a and Supplementary Fig. 4). Overall, the single-particle chamber occupancy ratio was found to be as high as 97% (Fig. 3b). The statistics of cell/bead occupancy rate and pairing rate are shown in Fig. 3c. A pairing rate of about 95% was achieved, which is a significant improvement compared to other scRNA-seq platforms. Finally, with a high speed flow of solution in the reverse direction, nearly 100% of the trapped barcoded beads could be recovered for downstream processing (Fig. 3c and Supplementary Fig. 5), which outperformed other platforms such as Drop-seq[13]. The combination of high loading rate, high pairing rate and remarkable recovery rate avoids loss of cell information.

In addition to the capacity of compartmentalization of single cells/beads with high efficiency, Paired-seq chip was designed to capture cells with minimum loss even with low-input cell number. Different low numbers (40, 80, 100, 200, 300, 400, 500, 800) of input cells were injected, and the capture efficiency (Fig. 3d) was calculated. The result showed that as high as 90% of

input cells could be captured. Such a high capture efficiency for a low input number of cells will be of great significance in dealing with precious cell samples.

**Cell-free RNAs removal capability.** Preparation of a single-cell suspension sample remains one of the most difficult tasks for scRNA-seq to generate meaningful biological representative data. It is difficult to identify the true composition of the original sample because of the presence of cell-free RNAs derived from tissue digestion and cell death. Paired-seq chip allows independent loading and washing of cells and beads independently which can prevent the barcoded beads from being contaminated by cell-free RNAs in the cell solution. To verify the capability of cell-free RNAs removal on Paired-seq platform, TAMRA fluorescent dye and PBS solutions were loaded into the cell capture channel and bead capture channel, respectively. The connection channel was kept blocked for 6 h, and there was no observable increase of fluorescence intensity in the bead capture channel (Supplementary Fig. 6A, B, Supplementary Movie 6), indicating the excellent isolation effect of the blocking valve to avoid contamination from

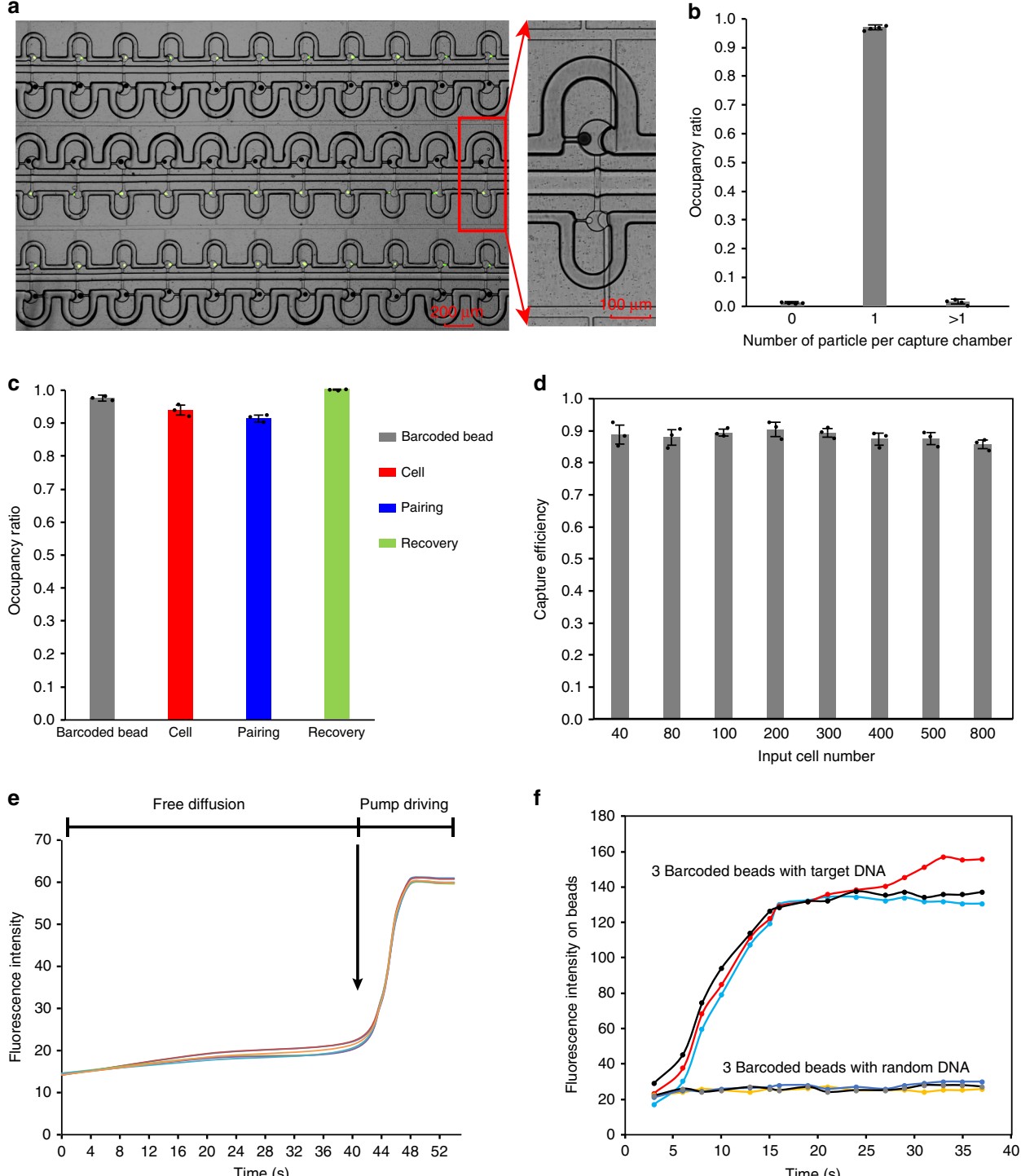

**Fig. 3 Profiling of Paired-seq chip. a** Image of Paired-seq chip loaded with single cells and single beads, which were compartmented in water-in-gas droplets. **b** The occupation ratio of single particle in Paired-seq chip. Error bars, mean ± s.d., $n = 4$. **c** The statistical chart of bead and cell occupation ratio, pairing ratio and bead recovery ratio. Error bars, mean ± s.d., $n = 3$. **d** Single cell capture efficiency with different numbers of input cells. Error bars, mean ± s.d., $n = 3$. **e** Change of cell chamber fluorescence intensity indication mixing efficiency of TAMRA dye solution in bead chamber with PBS in cell chamber under conditions of free diffusion and pump driving. **f** Characterization of DNA hybridization on the surface of barcoded beads with target DNA and random DNA. Source data are provided as a Source Data file.

cell-free RNAs during cell/bead solution loading. Considering the low sensitivity of fluorescence imaging, a small number of RNA molecules could also be amplified in the subsequent reactions, such as PCR amplification and sequencing, which would affect the experimental results seriously. Therefore, we also used the

sequencing method to further verify the isolation effect of the blocking valve and the cleaning effect. Total RNAs extracted from the same number of cells with a different species, considered as cell-free RNAs, were doped into human/mouse cell loading solution. Cells were captured in the chambers and washed with

1 × DPBS as the blocking valves were still activated. In neither test (mouse cells with human RNAs contamination or human cells with mouse RNAs contamination) did we detect obvious cell-free RNAs contamination from the other species (Supplementary Fig. 6C, D). Our results verified the complete isolation achieved by the blocking valve and the complete removal of background cell-free mRNAs in the cell suspension after washing, which is a significant advantage over other scRNA-seq platforms, such as Drop-seq and Seq-well.

**Rapid cell lysis and mRNA capture with active pumping**. Other challenges in the preparation of single-cell transcriptome sequencing samples, such as the lengthy time for single-cell lysis and poor mRNA capture efficiency, will affect the quality of the sequencing library and further affect the accuracy and sensitivity of the sequencing results. To evaluate the efficiency of mixing between paired droplets on our Paired-seq chip, a paired-droplets array was generated with one droplet containing TAMRA solution and PBS in the other. The blocking valve was then turned off to allow mixing of solutions in the cell and bead chambers. A very slow increase of fluoresce intensity was observed in the bead capture chamber due to free diffusion. In contrast, immediately after activating the two driving pumps, the fluorescence intensity in the cell capture chamber increased sharply and reached saturation in a few seconds, demonstrating rapid mixing between paired droplets enabled by the driving pumps (Fig. 3e, Supplementary Fig. 7, and Supplementary Movie 7). As a result, in the picoliter reactor, a cell can be completely lysed within 2 min (Supplementary Fig. 8 and Supplementary Movie 8) and FITC-labeled poly(A) DNA can be pumped from the cell chamber to the bead chamber and captured on beads within 20 s (Fig. 3f), indicating that the specific hybridization between mRNAs and barcoded bead can be performed rapidly on Paired-seq chip. To test the influence of shear forces on RNA quality or transcription, we compared the gene detection ability reflecting RNA integrity with different loading time. Our results suggested that at the loading time of 15 min and 40 min, the number of detected genes show no significant difference, suggesting that loading time does not cause spurious/stress to transcription (Supplementary Fig. 9A). We also analyzed the expression levels of 9 genes (ARF1, CAST, CDK7, DBI, DDIT3, ENO2, ETF1, PLOD2, and RGS2) reported to have correlations with mechanical stress[32]. Herein, the nine genes were biologically well characterized in terms of protein function, including cell communication, cell signaling, cell cycle, stress response and calcium release. There were no remarkable differences of the gene expression described above between the samples with different loading time, indicating that the shear force did no damage to the cells (Supplementary Fig. 9B). Controllable, rapid, and efficient cell lysis and mRNA capture in picoliter chambers enabled by active pumping is essential for high sequencing accuracy and sensitivity.

**Single-cell mRNA sequencing**. To assess the feasibility of scRNA-seq on this platform, we performed a mixed-species experiment with cultured human (K562) and mouse (3T3) cells, and the sequencing result of cell barcodes is shown in Supplementary Fig. 10. By avoiding the limiting dilution of cell and barcoded bead compartmentalization, 768 cell barcodes were successfully harvested with high quality in an 800-array Paired-seq chip, demonstrating very high efficiency on both the cells and the barcoded beads utilization (Fig. 4a). The result of the human-mouse experiment is shown in Fig. 4b, and each dot represents a cell barcode and number of UMIs derived from the human/mouse source. The closer the dots to the x-axis/y-axis, the higher purity of the cell barcode for the corresponding single species.

Among all the 768 harvested cell barcodes, 386 were identified as human species and 376 as mouse, yielding less than 0.8% mixed-species dots (while 2.4% mixed-species dots with 2000-unit Paired-seq chip)(Fig. 4b and Supplementary Fig. 2). Compared with other available scRNA-seq platforms, our Paired-seq showed a very low doublet rate (Supplementary Fig. 10B). The results established excellent single-cell integrity for scRNA-seq and indicate an obvious advantage in detection of transcripts and genes of Paired-seq.

To test the reproducibility of Paired-seq, different numbers of cells were harvested at different sequencing depths and culture times. We collected 188 and 248 K562 cells at an average sequencing depth of $18 \times 10^3$ and $39 \times 10^3$ mapped reads per cell, respectively. Technical replicates showed very high reproducibility (Pearson correlation, $R = 0.979$, Fig. 4c). In addition, our platform has the ability to evaluate the individual cell state according to cell-cycle scores, which were calculated for each human K562 cell based on previously reported phase-specific genes and methods[13]. Cells at different cell-cycle stages were clearly separated based on their cell-cycle scores (Fig. 4d). In general, Paired-seq presented high-efficient single-cell mRNA sequencing with reliable reproducibility and detection ability.

**Excellent sequencing accuracy and sensitivity with low cost**. Deeper sequencing depth can enhance the sensitivity of gene detection, but it can also significantly increase the cost. In order to balance sensitivity, accuracy, and cost, we analyzed the relationship between the sequencing depth and accuracy/sensitivity at single cell level of different platforms at the same time. To estimate the accuracy and sensitivity of Paired-seq, we compared the results with recent scRNA-seq platforms using ERCC and mES cells. About 100k molecules of ERCC and single mES cells were compartmentalized in the cell capture chamber and paired with individual barcoded beads to generate scRNA-seq libraries from ERCC and mES cells. In order to further optimize the operation on chip, we firstly compared the quality of sequencing data for ERCC experiment by using Paired-seq with enzymatic process on and off chip (in tube). The results showed that the percentages of mapped reads for enzymatic process on chip were significantly higher than that in tube. Most of the unmapped reads (due to too short sequence) in off-chip sample could be traced back to primer on the barcoded beads, which confirmed that the insufficient enzymatic reaction brought in technical noise (Supplementary Fig. 11). Then we followed the same protocol for subsequent sample preparation on Paired-seq chip. A total of 55 ERCC captured barcoded beads were sequenced at a depth of 1 million reads per bead. The ERCC sequencing data were processed with umis[33] software based on the existing benchmark. Additionally, 70 mES cells were sequenced at a saturated sequencing depth (over 0.5 million mapped reads per cell) and downsampled to a normalized depth of 0.5 million mapped reads per cell. Then the normalized data were randomly subsampled to reveal the corresponding changes in accuracy and sensitivity for each platform. The pipeline shown in Supplementary Fig. 12 was used to process the data for accuracy/sensitivity comparison with other platforms.

The accuracy here is defined as the Pearson product-moment correlation coefficient (R) between log-transformed detected ERCC counts and input ERCC spike-in counts per droplet which had been previously established[34]. The high accuracy achieved ($R = 0.973$) indicates that Paired-seq is a reliable platform to identify marker genes with low expression level (Fig. 4e and Supplementary Fig. 12c). ERCC sequencing data from Paired-seq were analyzed with established data analysis methods, with UMI merging and without merging, yielding capture efficiencies of 16.8% and 50%, respectively. Both values are slightly higher than

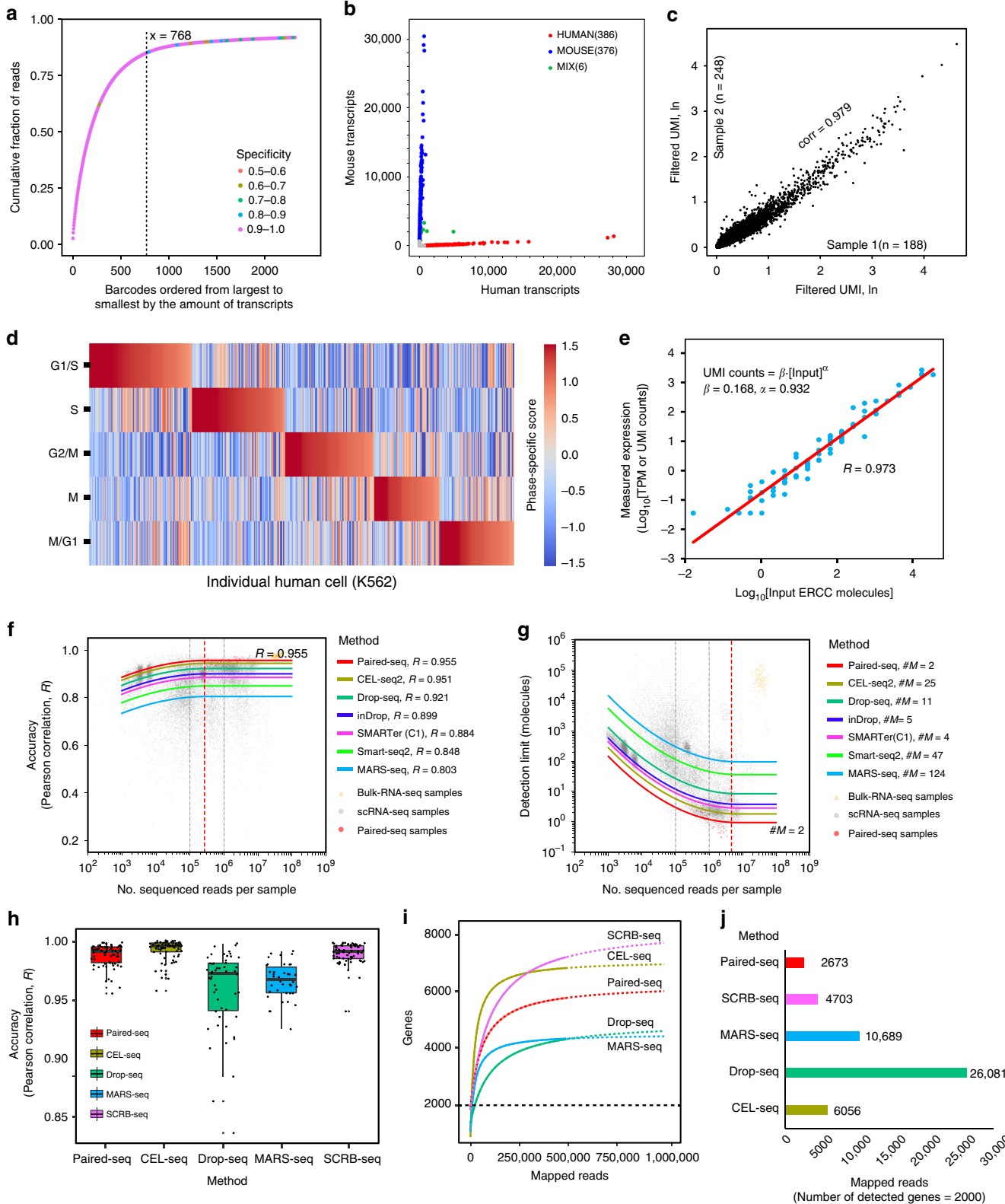

those of Drop-seq (12.8% and 47%), respectively[13]. Considering a global effect of sequencing depth, we also used a linear model, including an individual corrected performance parameter for each platform that could be ranked to account for the sequencing depth[34]. According to the model, we found that Paired-seq had the highest accuracy ($R = 0.955$) for ERCC detection (Fig. 4f) among 16 different kinds of scRNA-seq platforms (Fig. 4f and

Supplementary Figs. 13 and 15). In addition, we compared the accuracy of Paired-seq and a series of other scRNA-seq methods, including CEL-seq2/C1, Drop-seq, MARS-seq, and SCRB-seq, at the normalized sequencing depth (Fig. 4h) with data for mES cells. The Pearson correlation coefficient ($R$) of reference gene expression values for each cell and average expression of all the cells were calculated and shown in the plot by different

**Fig. 4 Characterization of Paired-seq platform. a** Plot of the cumulative fraction of reads vs barcodes accumulates which are arranged in decreasing order of size (number of transcripts) for the human-mouse mixture experiment. **b** Human-mouse mixture experiment using Paired-seq. **c** The technical repeats from two independent experiments indicate high correlation of 0.979. **d** Cell-cycle state of K562 was measured by Paired-seq. The cells were ordered by their phase scores. **e** Accuracy and mRNA capture efficiency evaluated by ERCC sample. **f, g** Models of accuracy and sensitivity with a global dependency on sequencing depth. Each model has 26 parameters and is fitted to $n = 20,772$ samples. Bulk data (pink triangles) are displayed only for context. Solid curves show the predicted dependence on sequencing depth. **f** Accuracy is only marginally dependent on sequencing depth. Saturation occurs at 270,000 reads per cell in the model (dashed red line). Methods are ordered by performance on the basis of predicted correlation (R) at 1 million reads. **g** Sensitivity is critically dependent on sequencing depth. Saturation occurs at 4.6 million reads per cell (dashed red line). The gain from 1 to 4 million reads per sample is marginal, whereas moving from 100,000 reads to 1 million reads corresponds to an order-of-magnitude gain in sensitivity (dashed black lines). Methods are ordered by performance on the basis of predicted detection limit (#M, number of molecules at 1 million reads). **h** Accuracy of single-cell resolution for mES cells. Each dot represents a cell and each box represents the median and first and third quartiles per replicate and method. 72, 77, 53, 38, and 70 cells were used for Paired-seq, CEL-seq, Drop-seq, MARS-seq and SCRB-seq. **i** Fitted (solid line) and predicted (dashed line) curve of median genes detected for single mES cell to varying mapped reads according to experiment results of five different platforms. Two-tailed F-test was performed to generate P-value to assess the accuracy of the curve (P-value > 0.05). **j** The number of mapped reads for five different platforms when the detected number of genes were 2000. Source data are provided as a Source Data file.

methods[15]. Paired-seq showed a very high accuracy ($R = 0.991$) possibly due to the small reaction volume, high mRNA capture efficiency and noises reduction with on-chip enzymatic operation. The results indicated that Paired-seq possessed superior performances in quantification of transcripts in single cells.

Similarly, the sensitivity was compared with other platforms using ERCC and data for mES cells (Supplementary Fig. 14)[34]. According to the logistic regression model[34] with ERCC, we achieved the detection limit of such as "10 molecules" for Drop-seq, "5 molecules" for inDrop, and also "2 molecules" for Paired-seq (Fig. 4g, Supplementary Fig. 15). The algorithm provided a fair comparison and demonstrated that sensitivity of Paired-seq was comparable to other modern single-cell approaches. In addition, in data processing for mES cells, we downsampled reads with normalized depth of each cell to varying lower mapped reads for each method, and drew the fitted curve of median genes detected for single mES cell versus different mapped reads (Fig. 4I). Although conventional methods, including CEL-seq2/ C1 and SCRB-seq, have higher gene detection ability due to the use of liquid barcoding primers, the large reagent consumption greatly increases the cost and the complexity of manual operations, thus it is unpractical for high-throughput single-cell analysis. Compared with the high-throughput scRNA-seq platform (Drop-seq), Paired-seq detected more genes per cell than Drop-seq at different sequencing depths, which may be attributed to the effective mixing of reagents and enzymatic process on Paired-seq chip and the smaller reaction chamber. Most importantly, based on the analyses of the sequencing depth (mapped reads per cell) vs. the number of detected genes of mES cells for five different sequencing platforms, only 2673 mapped reads were needed for Paired-seq which was the lowest compared with others when the number of detected genes is 2000 (Fig. 4i, j). This result shows that the cost of sequencing for Paired-seq is the lowest.

**Heterogeneous cellular subpopulations**. ScRNA-seq is a promising technology to identify and describe cellular subpopulations from heterogeneous populations of cells. ES cells are derived from a stage in which key early lineage specification events are occurring. Specifically, upon Leukemia Inhibitory Factor (LIF) withdrawal, ES cells will experience unguided differentiation and generate various subpopulations[35]. Compared to fully differentiated cell types, ES cells in serum are relatively homogeneous, with only some well-characterized fluctuations even in a short time after LIF withdrawal. Study of heterogeneity information from such relatively homogeneous cell populations poses a challenge for single cell sequencing. For further verification, the ability to distinguish such relatively weak heterogeneity by

Paired-seq, mES cells were collected and analyzed in nine batches over 10 days after LIF withdrawal (Supplementary Figs. 16 and 17). Upon LIF withdrawal, the time series samples collected at Day 0, 2, 4, 7, and 10, were assayed for the single-cell transcriptomes using Paired-seq. Replicate experiments were performed by different people on a few of the time points of this study. In the comparison of the biological replicates (Day0_1/2, Day7_1/2 and Day10_1/2), Paired-seq data makes their t-Distributed Stochastic Neighbor Embedding (t-SNE) points go together (Fig. 5a), suggesting the similar expression profiles of these replicates.

Overall, the combined single-cell expression profiles of these time points give five predominant cell clusters, which were readily correlated to the post-LIF times (Fig. 5b). The pseudo-time algorithm plots the trajectory that is concordant to the order of the sampling time (Supplementary Fig. 18). Some of the clusters enrich the markers of the differentiated cell types of expectation, such as *Cytokeratin* and *Otx2* etc., and reflects the fluctuation of pluripotency factors, *Zfp42*, *Pou5f1* and *Sox2* etc., which validate the capability of Paired-seq[36]. (Fig. 5c, d). In addition to those well-known transcription factor and markers, 1594 genes of fluctuated expressions were identified. These genes were differentially expressed ($p$-value < 0.05, Supplementary Table S3) among the five cell populations. Kyoto Encyclopedia of Genes and Genomes (KEGG) pathway enrichment analysis (Fig. 5e) and Gene Ontology (GO) enrichment analysis for biological process (Fig. 5f) revealed that these genes were mainly involved in some fundamental biological processes and pathways during cell differentiation ($p$-value < 0.05). In summary, Paired-seq is able to track down the population development and detect the fluctuated expressions of the key markers in the differentiation process. This is concordant to what has been described in inDrop[12].

**Heterogeneity of drug treated cancer cells**. To characterize the drug resistance and disease recurrence after anti-cancer treatments, Paired-seq was used to study the heterogeneity of anti-cancer drug treated cancer cells. *Nocodazole*, an antineoplastic agent and known as a cell cycle inhibitor that inhibits polymerization of microtubules[37], and possibly influences the differential mRNA transcription related to cell cycle, was used as a model drug to study the drug response at single cell drug to study the drug response at single cell level. ScRNA-seq samples of K562 cells before and after drug treatment were processed on Paired-seq chip and the sequencing data were analyzed by t-SNE (Supplementary Figs. 19 and 20). Default unsupervised clustering on Seurat[38] gives two putative clusters, which are readily associated to the treatment response, indicating an obvious phenotypic variability in response to the drug (Fig. 6A, B). As we can see, the

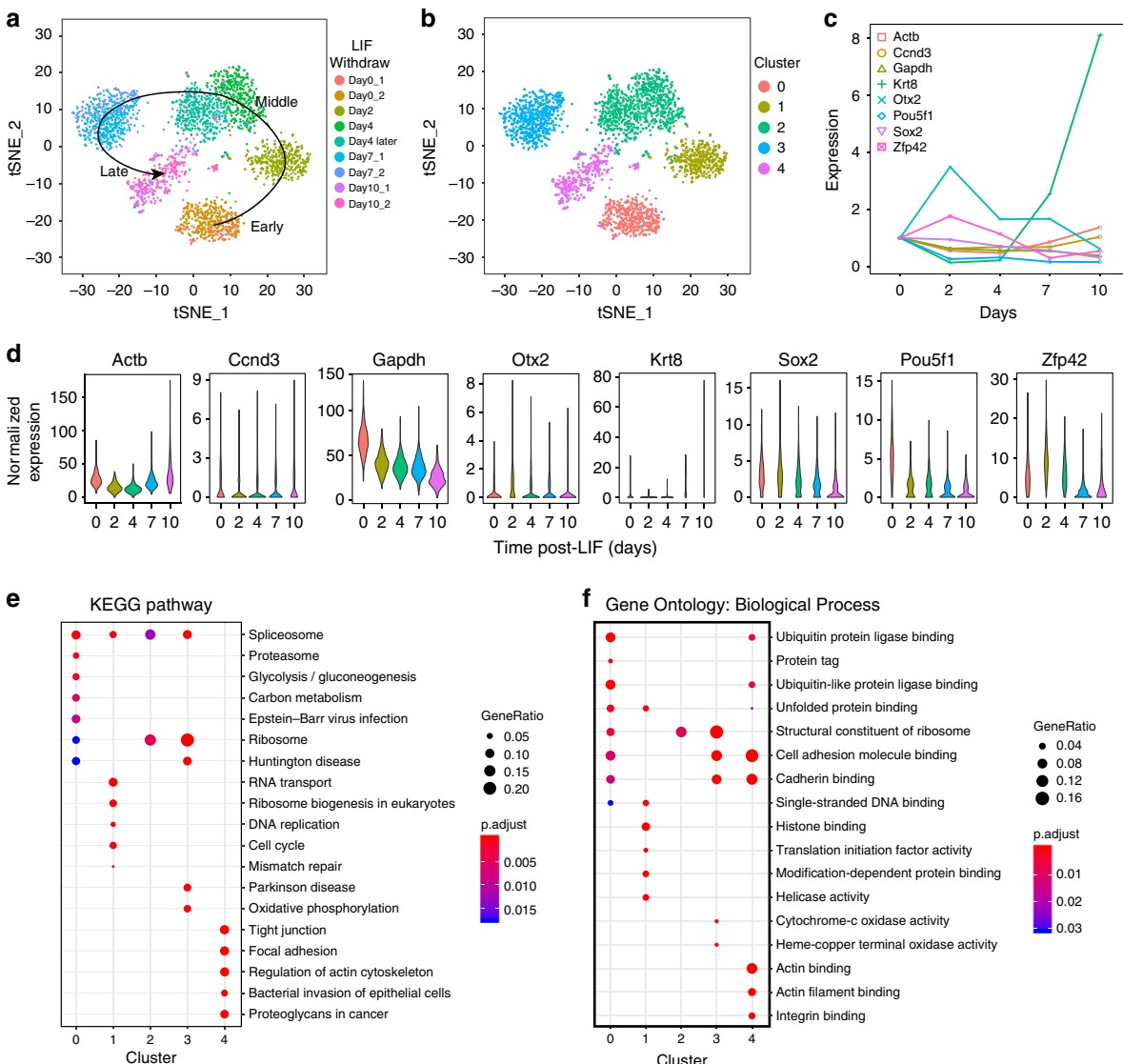

**Fig. 5 Heterogeneity of differentiated mouse ES cells. a** t-SNE maps of mES samples from different days after LIF withdrawal. Different experimental batches are labeled with different colors. **b** t-SNE maps of mES samples by unsupervised clustering ID. Five distinct clusters are labeled with different colors. **c** Average and **d** distribution of key pluripotent factors and differentiated markers of different time points after LIF withdrawal. **e** KEGG pathway enrichment analysis for five clusters. **f** Biological process analysis of gene ontology enrichment for five clusters Source data are provided as a Source Data file.

untreated replicates share the same clusters whereas the treated sample goes to a relatively segregated cluster indicating the phenotypic change. We characterize total 1179 differential expression genes (DEG) across the treatment conditions. Figure 6c, d shows GO term enrichment analysis of the top 50 genes that are elevated in the treated cluster. These terms guide us to the genes that are correlated to the activated mitosis, such as *ASPM, AURKA, CENPF, KIF208, TOP1, RNF8, SEPT7, SMC3, TPX2,* and *CENPE*, which validates the accuracy of our single-cell RNA-seq assay (Fig. 6e). All of these results consistent with previous knowledge[39] proved the heterogeneity of cancer cells in anti-cancer ability and drug resistance. The results show that Paired-seq can provide comprehensive genetic expression analysis of individual cells to reveal the heterogeneity in anti-cancer drug responses, thereby facilitating the development of optimized clinical anti-cancer strategies.

## Discussion

In summary, we proposed a high-throughput single-cell RNA sequencing platform named Paired-seq with high cells/beads

utilization efficiency, cell-free RNAs removal capability, high gene detection ability, and low cost. By using the differential flow resistance principle, Paired-seq overcomes the waste of reagents and loss of cells caused by traditional limiting dilution methods and achieved utilization efficiency up to 95% in both single cell/ barcoded bead isolation and pairing. High-efficiency single cell/ bead paring is a promising technology for analysis of precious and rare cells, such as stem cells, neuron cells, or CTCs. Furthermore, Paired-seq allows real-time observation of single cells that cannot be available in droplet-based platforms like Drop-seq or InDrop. Integration of controllable valves and pumps enables complete removal of cell-free RNAs, efficient cell lysis and mRNA capture. The clear background without cell-free RNAs helps to eliminate noise and faithfully reflects the true composition of the original sample. The efficient cell lysis and mRNA capture endow the highest mRNA detection accuracy ($R = 0.955$) and comparable sensitivity compared to other scRNA-seq platforms. The high gene detection capacity allows lower sequencing costs because it requires less sequencing depth to achieve the same number of detected genes. By using Paired-seq to investigate the

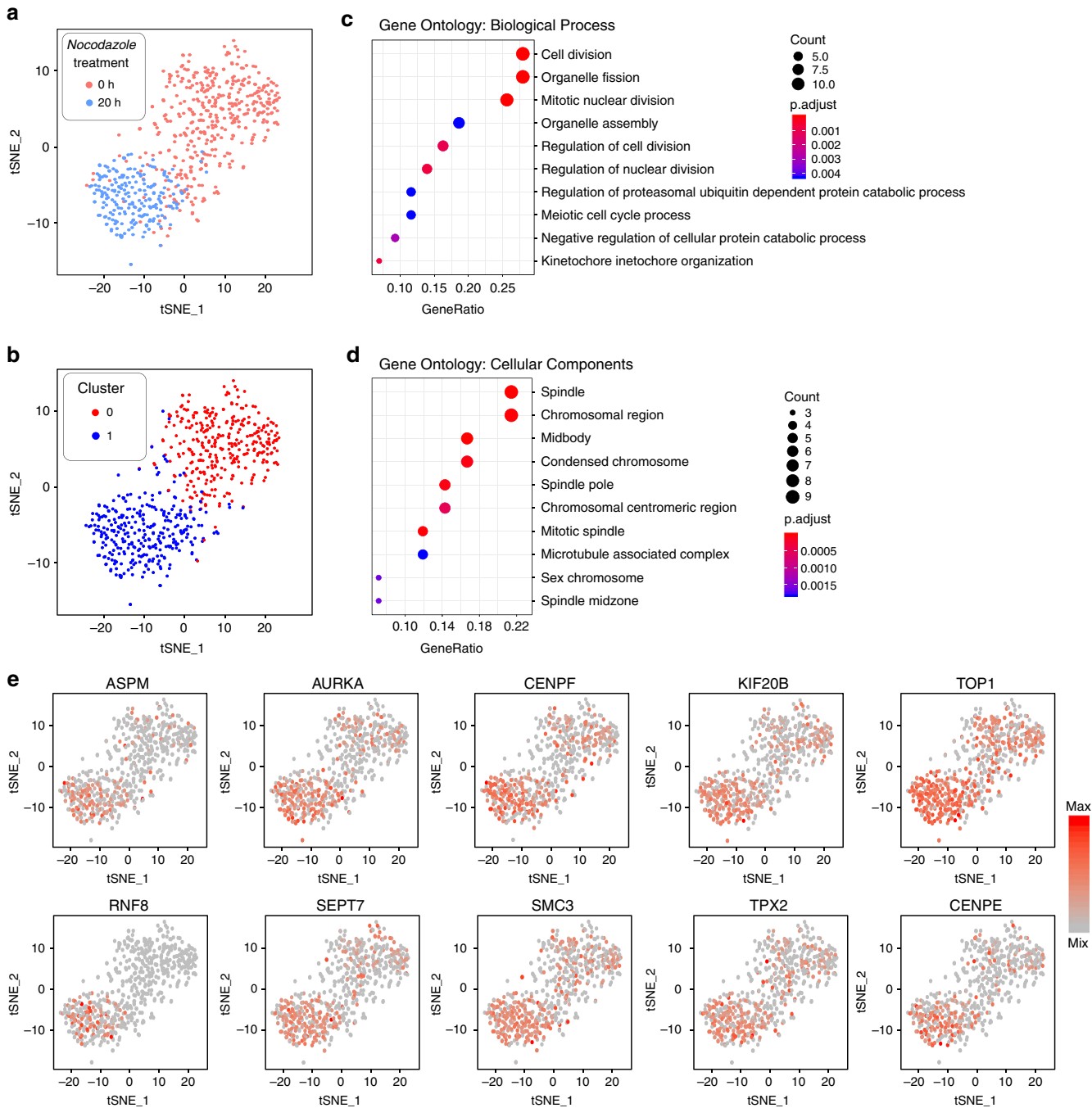

**Fig. 6 Heterogeneity of *Nocodazole* treated cancer cells. a** t-SNE visualization of K562 cells (RNA-seq) colored by *Nocodazole* treatment time or **b** unsupervised clustering ID. **c** Biological process and **d** cellular components analysis of gene ontology enrichment for the top 50 genes that are elevated in the treated cluster. **e** t-SNE maps of *Nocodazole* treated and untreated K562 cells. Gene expression levels are indicated by shades of red. All the genes shown in the map are correlated to the activated mitosis. Source data are provided as a Source Data file.

single-cell expression profile of mES cells and anti-cancer drug treated cells, we verified the reproducibility and significant detection ability for varying genetic expression, presenting great potential for cell biology, developmental biology and precision medicine.

## Methods

**Chip fabrication.** The silicon mold for single cells and single beads manipulation was fabricated by conventional photolithography. Mask fabrication with twice overlay exposure was applied to produce three different heights of the flow layer. First, SU-8 3010 photo-resist (MicroChem) was coated on a silica wafer to produce the connection channel with 8 μm height. Then, GM 1070 photo-resist (Gersteltec)

was coated on the same wafer to produce the cell capture channel with 30 μm height. Next, the micro-sphere capture channel was produced with SU8 3050 photo-resist (MicroChem). The control layer was fabricated with one step exposure using GM 1070 photo-resist. Finally, the mask was coated with 0.7% 1 H, 1 H, 2 H, 2H-perfluorooctyldimethyl-chlorosilane/GH-135 (v/v) solution and dried to make the surface hydrophobic. The silica wafer with flow layer pattern was placed in a 60 mm plastic petri dish and PDMS precursor solution (10:1 of polydimethylsiloxane and curing agent) was poured on the silica wafer and cured at 75 °C for 10 min to the 80% repolymerization. The silicon wafer with control layer pattern was spun with PDMS precursor solution (23:1 of polydimethylsiloxane and curing agent) and then put on a horizontal heater at 48 °C for 7.5 min to the 80% repolymerization. The flow layer and the control layer were perfectly aligned under the microscope, and bound at 48 °C for 25 min for the complete bond between two layers. Then the PDMS with flow layer and control layer pattern was peeled off,

and punched for the inlets and outlets with a 1.0 mm puncher. The final chip was fabricated by bonding the integrated PDMS with a 2.5 mm × 7.5 mm glass immediately after oxygen plasma treatment.

**Cell culture and preparation.** Human K562 cells (ATCC CCL-243, purchased from National Infrastructure of Cell Line Resource) and mouse 3T3 cells (ATCC CRL-1658, purchased from National Infrastructure of Cell Line Resource) were cultured in Dulbecco's Modified Eagle Medium (DMEM, ThermoFisher) supplemented with 10% Fetal Bovine Serum (FBS, ThermoFisher) and 1% penicillin-streptomycin (ThermoFisher) at 37 °C and 5% $CO_2$. 3T3 cells were harvested by 0.25% trypsin-EDTA (Life Technologies) and re-suspended with 1 mL 1× DPBS in a 1.5 mL centrifuge tube. Unlike 3T3 adherent cells, suspended K562 cells were pipetted up and down gently several times and then directly pipetted out, followed by centrifugation and suspension with 1 mL 1× DPBS in a 1.5 mL centrifuge tube. For mixed-species experiments, human K562 cells and mouse 3T3 cells were mixed in a 1:1 ratio with a final concentration of 0.2% Poloxamer 188 (F68, Thermo-Fisher) and 4% FBS in 1× DPBS buffer.

The mouse embryonic stem (mES) cells were J1 mouse embryonic stem cell (J1 mES cell): derived from the mouse 129 s4/SvJae strains called J1. They were kindly provided by Stem Cell Bank, Chinese Academy of Sciences. The culture flasks were pre-treated with gelatin at 37 °C and 5% $CO_2$. For the undifferentiated stage, mES cells were cultured in DMEM supplemented with 15% FBS, 2 mM L-glutamine, 0.1 mM non-essential amino acids (NEAA), 0.1 mM 2-mercaptoethnol and 1000 U mL$^{-1}$ LIF. LIF was removed for unguided mES differentiation. We collected mES cells on the 0th, 2nd, 4th, 7th, 10th days after LIF withdrawal for subsequent experiments. Before injecting into the chip, the mES cells were washed and re-suspended in 1× DPBS with a final concentration of 0.2% F68 and 4% FBS.

**Preparation of barcoded.** Commercial barcoded beads were purchased from ChemGenes Company (Wilmington, Massachusetts, USA; cat. Macosko-2011-10 (V+)) described in Drop-seq[13]. The oligo synthesis scale was 10 μmole. Subsequently commercial barcoded beads were washed twice with 30 mL of TE/TW (10 mM Tris pH 8.0, 1 mM EDTA, 0.01% Tween), re-suspended in 10 mL TE/TW and passed through a 40 μm strainer (PluriSelect) into a 50 mL Falcon tube. Then they were placed at 4 °C for long-term storage. Before experiments, 1000 barcoded beads were used and re-suspended in 10 μL 2% sodium alga acid solution with 0.2% Triton X-100 for subsequent capture in the chip.

**Paired-seq operation.** All aqueous suspensions were loaded into 1 mL plastic syringes. The blocking valve was turned on to disconnect the cell and bead chambers, resulting in no fluid exchange between them. Barcoded beads suspended in 2% sodium alga acid and 0.2% Triton X-100 were injected into the chip through the bead inlet at a flow rate of 0.2 mL h$^{-1}$, while 1 x DPBS was injected through the cell inlet buffer inlet at a flow rate of 0.06 mL h$^{-1}$. After finishing the bead capture, the bead channel was washed with 1× DPBS to replace sodium alga acid and Triton X-100 while the driving pump for bead was activated to prevent bead escape. Then, the cell suspension and DPBS buffer were respectively injected into the chip through the cell inlet at 0.015 mL h$^{-1}$, with the speed of 0.03 mL h$^{-1}$ of 1× DPBS in the bead channel. After finishing single cells capture, the cell driving pump for cell was pressure-forced to prevent the cells from escaping, and then the cell channel was washed with 1× DPBS to remove residual cells in the channel. Next, the buffer in the bead channel was replaced with lysis buffer (160 mM Tris pH 7.5 (ThermoFisher), 0.16% Sarkosyl (Sigma), 16 mM EDTA, 0.5 U μL$^{-1}$ RNase Inhibitor (TransGen Biotech), 0.12% F68). Then the cell and bead inlets were unplugged and both bead and cell outlets were blocked. Gas was reversely injected into the channels, generating water-in-gas droplets, which contained single beads and single cells. Then the blocking valve was turned off to enable solution exchange between the paired chambers. The whole procedure could be real-time monitored to ensure complete lysis of cells. At the same time, the released mRNA molecules were captured by the paired beads. By alternately activating the driving pump for cells and the driving pump for beads, solutions in two chambers could be easily transferred back and forth, thus allowing efficient cell lysis and mRNA capture. After turning on the blocking valve, the cell channel and bead channel were washed with 1× DPBS independently. The driving pump for barcoded beads was activated to keep the trapping of mRNA captured beads, and the reverse transcription mix (1x RT buffer (Fermentas), 1 mM dNTPs (TransGen Biotech), 1 U μL$^{-1}$ RNase Inhibitor, 2.5 μM Template_Switch_Oligo (Life Technologies), and 10 U μL$^{-1}$ Maxima H-RT (Fermentas)) was injected in both channels. The chip was incubated at room temperature for 30 min followed by 42 °C for 90 min.

After reverse transcription, the beads were washed with TE-SDS (10 mM Tris pH 8.0, 1 mM EDTA, 0.5% Sodium Dodecyl Sulfate (Sigma)), 20 μL TE/TW, and 20 μL TE (10 mM Tris pH 8.0), with driving pump activated to trap the barcoded beads in the original position. Then 20 μL Exonuclease I mix (1x Exonuclease I Buffer and 1 U μL$^{-1}$ Exonuclease I (NEB)) was injected into the chip to remove the excess primers by incubating the chip at 37 °C for 45 min.

The channels were then washed with 10 μL TE/SDS, 10 μL TE/TW, 10 μL ddH$_2$O to remove Exonuclease I mix. After reducing the pressure of the bead driving pump, a high speed of solution was introduced to push the advance of beads, making them gather at the end of channel. With the help of water phase flow

and gas phase flow in the direction of bead outlet, the barcoded beads could be collected from outlet into tubes without remnant.

**Feasibility testing.** Total RNAs were extracted from human K562 and mouse 3T3 cells by using GeneJET RNA Purification Kit (Thermo Fisher) according to the manuals and protocols. The products were quantified by NanoDrop ND-2000. The RNAs released by 10$^6$ 3T3 cells were mixed with 10$^6$ K562 cells and injected into the chip through the cell inlet, while the same amount of cell-free RNAs of K562 cells mixed with 10$^6$ 3T3 cells were injected into another chip. Cell capture continued 30 min, and then the cell channel was washed by 1× DPBS. The cleaning process was also set at 30 min. Subsequent manipulation was the same as the normal process. The sequencing data were aligned to hg19_mm10 to test the presence of cell-free mRNA information, which verified the capacity of the blocking valve and cleaning effect.

**ERCC experiment.** External RNAs (ERCC RNA Spike-In Mix) were purchased from ThermoFisher. The originating ERCCs were diluted to $1.2 \times 10^9$ μL$^{-1}$ with 1x PBS + 1 U μL$^{-1}$ RNase Inhibitor (Lucigen). After processing the ERCCs in a Paired-seq chip, the theoretical number of ERCCs contained in each cell chamber was about 10$^5$ molecules. In order to reduce low quality of ERCC reads by STAR, sequencing reads were aligned to a dual ERCC-human reference, where human sequences were used as "bait".

**cDNA amplification and library preparation.** All the collected beads were aliquoted into one PCR tube for PCR amplification. The PCR program was as follows: 95 °C for 3 min; and then four cycles of: 98 °C for 20 s, 65 °C for 45 s, 72 °C for 3 min; then 10 cycles of 98 °C for 20 s, 67 °C for 20 s, 72 °C for 3 min; then a final extension step of 5 min. The PCR products were purified using 0.6x VAHTS DNA Clean Beads (Vazyme Biotech) according to the manual twice, and eluted in 11 μL H$_2$O. The concentration of the purified products was quantified by qubit3.0.

The 3'-end enriched sequencing library was prepared using a TruePrep DNA Library Prep Kit V2 for Illumina (Vazyme Biotech), according to the manufacturer's instructions, except that the custom primer P5 was used in place of the kit's oligos. The samples were then amplified as follows: 72 °C for 3 min, 98 °C for 30 s; and 12 cycles of: 98 °C for 15 s, 55 °C for 30 s, 72 °C for 30 s; then a final extension step of 5 min. The 3'-end enriched library products were purified using 0.6x VAHTS DNA Clean Beads (Vazyme Biotech), and eluted in 11 μL H$_2$O. The concentration was quantified by qubit3.0. The fragment size of the 3'-end enriched sequencing library was analyzed by Qsep-100, and the average size was between 450 and 650 bp. The libraries were sequenced on the Illumina Nextseq 550 according to the manufacturer's instructions, except that Custom read 1 was used for priming of read 1. Read 1 was 21 bp; read 2 was 60 bp for all the experiments.

**Single-cell responses to Nocodazole.** Nocodazole was purchased from Selleck (#S2775) and dissolved in DMSO at the concentration of 2 nM. Before the experiment, K562 cells were cultured in DMEM supplemented with 10% Fetal Bovine Serum, 1% Penicillin-streptomycin and 1 nM *Nocodazole* for 20 h. For flow cytometry analysis, the medium containing *Nocodazole* was removed, and the cells were re-suspended in 400 μL cold 1× PBS and 1100 μL cold fixing solution (100% ethyl alcohol) and stored overnight at 4 °C. The next day cells were centrifuged to remove the fixing solution, washed three times with 1× PBS and re-suspended in 500 μL 1× PBS. RNase A (ThermoFisher, 20 mg L$^{-1}$) was introduced to remove the interference of RNA at 37 °C for 1 h. Then the nuclear DNAs of K562 cells were stained by PI (ThermoFisher, 50 mg L$^{-1}$) in a dark place at 4 °C for 1 h. One million K562 cells in total were detected by flow cytometry, with the obvious fluorescence peak corresponding to 2n/4n DNAs, which represented the cell cycle position. For Paired-seq, after removing the Nocodazole, K562 cells were washed three times and re-suspended in DMEM supplemented with 0.2% F68 and 4% FBS.

**Data sources.** Raw read data from published studies were downloaded from either ENA or SRA, as listed in Supplementary Table 5. We followed the same protocol of the data analysis and confirmed with the authors about the details of the data analysis process[33].

**Data analysis workflow for ERCC sample.** ERCC sequencing data prepared on the Paired-seq platform was processed with umis workflow to obtain a digital expression matrix for performance estimation and comparison. The raw sequencing data fastq files were first transformed to a single fastq file with "UMIs fastq transform" using Drop-seq mode. Then pseudo-alignment was performed with Rapmap to obtain the sam file. After that, a digital expression matrix was produced by "UMIs tagcount". For bulk RNA-seq and other scRNA-seq platforms, we used the processed data provided by Svensson et al.[33].

For each individual cell or sample, specific ERCC spike-in molecules were proved to be detected with at least one copy observed. After discarding the undetected ERCC spike-in types, we calculated the Pearson correlation coefficient (R) between detected ERCC counts (UMI) and input ERCC counts from the

equation below as the accuracy of each sample:

$$\log_{10}(\mathbf{UMI_i}) = \alpha \cdot \log_{10}(\mathbf{Input_i}) + c. \tag{1}$$

We can get the UMI efficiency (mRNA capture efficiency) of UMI based protocols at the same time, namely $\beta$ in the following equation:

$$\mathbf{UMI_i} = \beta \cdot [\mathbf{Input}]^{\alpha}. \tag{2}$$

When we model the relation between read depth and performance metrics for individual protocols, we use a linear model with a quadratic term for read depth to capture diminishing returns on investment. The model considers the read depth effect to be global, and has a categorical performance parameter for each protocol:

$$\mathbf{metric} = a^2 \cdot \log_{10}(\mathbf{reads_i}) + b \cdot \log_{10}(\mathbf{reads_i}) + \text{performance}_{\text{protocol}} + \varepsilon. \tag{3}$$

Here the performance metric will plateau and saturate when

$$\log_{10}(\mathbf{reads_i}) = -\frac{b}{2a}. \tag{4}$$

For sensitivity calculation, we transformed the detected spike-in count into a binary variable (detected (1) or undetected (0)). Then we built a logistic regression model with Python scikit-learn package for each sample:

$$\mathbf{p}(\mathbf{detected_i}) = \frac{1}{1 + e^{-(a \times \log(M_i) + b)}} + \varepsilon. \tag{5}$$

The sensitivity was calculated as the molecule count when the detection probability equals to 0.5, namely:

$$\text{detection limit} = -\frac{b}{a}. \tag{6}$$

**Processing of the Paired-seq data**. For all the sequencing results, each dataset was generated from one single chip. One single experiment was pooled to generated one data. Paired-seq sequencing libraries produce paired-end reads: Read 1 contained a cell barcode (12 bases) and a UMI (8 bases); Read 2 contained mRNA information. The reads would be preprocessed with the following steps, correcting bead, filtering low-quality reads, trimming read 2 including polyA, adapter and primer, alignment, assigning gene tags, generating digital gene expressing. The beads with the twenty-first base as A, C or G only were used in our experiments. If the number of continuous T bases at the end of read 1 was less than or equal to 12, we inserted "N" bases before T bases. Otherwise, the pair of reads was dropped. Filtering low-quality reads was based on the base quality of the cell barcode and UMI. Respectively, cell barcode and UMI should have only one base with quality lower than 20 at most. Otherwise, the read pair was discarded. At least 5 contiguous bases of TSO and at least 6 contiguous bases of A with no mismatch were examined for read 2 and were hard clipped off the read. At least 6 contiguous bases of primer with one mismatch allowed considered for read 2 and hard clipped off read. The read pair was discarded, if the length of read 2 was less than 26 after trimmed. STAR alignment tool was used to align read 2 with the reference genome. For human and mouse mixed cells, we used hg19_mm10 mentioned in Drop-seq as reference genome. This program from Drop-seq added a tag "GE". We kept the unique mapping with gene tags. Then unique UMIs for each gene of each cell were counted to generate digital gene expression.

**Theory supplement**. Using the Darcy–Weisbach equation to determine pressure difference in a microchannel and solve the continuity and momentum equations for the Hagen–Poiseuille flow problem, we obtained the pressure difference $\Delta P = fL\rho V^2/2D$, where $f$ is the Darcy friction factor, $L$ is the length of the channel, $\rho$ is the fluid density, $V$ is the average velocity of the fluid, and $D$ is the hydraulic diameter, respectively. $D$ can be further expressed as $4A/R$ for a rectangular channel, and $V$ as $Q/A$, where $A$ and $R$ are the cross-sectional area and perimeter of the channel, and $Q$ is the volumetric flow rate. The Darcy friction factor, $f$, is related to aspect ratio, $\alpha$, and Reynolds number, $\text{Re} = \rho VD/\mu$, where $\mu$ is the fluid viscosity. The aspect ratio is defined as either height/width or width/height such that $0 \leq \alpha \leq 1$. The product of the Darcy friction factor and Reynolds number is a constant that depends on the aspect ratio, i.e., $f \times \text{Re} = C(\alpha)$, where $C(\alpha)$ denotes a constant that is a function of the aspect ratio, $\alpha$. After simplifications, by applying the Darcy–Weisbach equation to a rectangular channel, we obtain the expression:

$$\Delta P = \frac{C(\alpha)}{32} \frac{\mu LQR^2}{mA^3}. \tag{7}$$

In the simplified circuit diagram of the trap., fluid can flow from junction a to b (c to d) via path 1 or 3 (path 2 or 4) (Supplementary Fig. 21). Ignoring minor losses due to bends, widening/narrowing, etc., Eq. (7) is applied separately for paths 1 and 3 (path 2 or 4), and because the pressure drop is the same for both paths, we equate both expressions to yield

$$\frac{Q_1}{Q_3} = \left(\frac{C_3(\alpha_3)}{C_1(\alpha_1)}\right) \cdot \left(\frac{L_3}{L_1}\right) \cdot \left(\frac{R_3}{R_1}\right)^2 \cdot \left(\frac{A_1}{A_3}\right)^3 (\text{a to b}) \tag{8}$$

and

$$\frac{Q_2}{Q_4} = \left(\frac{C_4(\alpha_4)}{C_2(\alpha_2)}\right) \cdot \left(\frac{L_4}{L_2}\right) \cdot \left(\frac{R_4}{R_2}\right)^2 \cdot \left(\frac{A_2}{A_4}\right)^3 (\text{c to d}), \tag{9}$$

where subscripts 1 and 3 denote paths 1 and 3, respectively. For path 1, the length, L1, is assumed to be that of the narrow channel to simplify analysis. This is valid because most of the pressure drop occurs along the narrow channel. For the trap to work, the volumetric flow rate along path 1 must be greater than that of path 3, i.e., Q1 > Q3. Using the relationships $A = W \times H$ and $P = 2\bullet(W + H)$, where $H$ is the height of the channel, we arrive at

$$\frac{Q_1}{Q_3} = \left(\frac{C(\alpha_3)}{C(\alpha_1)}\right) \cdot \left(\frac{L_3}{L_1}\right) \cdot \left(\frac{W_3 + H_C}{W_1 + H_C}\right)^2 \cdot \left(\frac{W_1}{W_3}\right)^3 > 1 \tag{10}$$

and

$$\frac{Q_2}{Q_4} = \left(\frac{C(\alpha_4)}{C(\alpha_2)}\right) \cdot \left(\frac{L_4}{L_2}\right) \cdot \left(\frac{W_4 + H_B}{W_2 + H_B}\right)^2 \cdot \left(\frac{W_2}{W_4}\right)^3 > 1, \tag{11}$$

$$C_{(\alpha)} = f \times Re = 96 \times \big(1 - 1.3553 \cdot \alpha + 1.9467 \cdot \alpha^2 - 1.7012 \cdot \alpha^3 + 0.9564 \cdot \alpha^4 - 0.2537 \cdot \alpha^5\big). \tag{12}$$

Note that this final expression does not contain any fluid velocity term, implying that a properly designed trap will work for all velocities in the laminar flow regime.

**Reporting summary**. Further information on research design is available in the Nature Research Reporting Summary linked to this article.

## Data availability
The sequencing data presented in this paper have been deposited in the Sequence Read Archive (SRA) under BioProject accession number PRJNA578456 [https://trace.ncbi.nlm.nih.gov/Traces/sra/?study=SRP226387]. SRA, PRJNA305381; GEO: GSE75790, etc. were referenced in the (supplementary dataset) manuscript. Source data are available in the Source Data file. All other data are available from the authors upon reasonable request.

## Code availability
The same data processing packages were used as Dropseq[13] to analyze the sequencing data. The packages can be found at https://github.com/broadinstitute/Drop-seq/releases.

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

## Acknowledgements

The authors thank the National Science Foundation of China (21927806, 21735004, 21521004, 21325522), the National Key R&D Program of China (2018YFC1602900), Innovative research team of high-level local universities in Shanghai, and the Program for Changjiang Scholars and Innovative Research Team in University (IRT13036) for their financial support.

## Author contributions

Chaoyong Yang conceived the project. Mingxia Zhang designed the microfluidic device. Yuan Zou and Xing Xu designed the biological experiments and developed the scRNA-seq protocol. Qin Chen carried out the cDNA amplification and library preparation experiments. Chaoyong Yang, Richard N. Zare, Wei Lin and Zhi Zhu supervised the research. Mingxuan Gao, Xuebing Zhang and Jia Song performed mRNA-seq data analysis. All authors proofread the manuscript and provided comments.

## Competing interests

The authors declare no competing interests.
