## [Peer Review File · Nature Communications]

Reviewers' Comments:

Reviewer #1:

Remarks to the Author:

Zhang et al and colleagues describe a novel microfluidics platform enabling single cell sequencing with very minimal cell loss. The crux of the approach lies in efficient control of cell and bead loading into individual chambers using differential flow resistance. This new method results in a near perfect co-encapsulation ratio 1:1 (beads:cells) while maintaining the doublet rate at a very low level (<1%). To achieve a comparable doublet rate, a typical Poisson loading would result in ~90% of cell and bead loss.

Beside preserving most of the cell input, this system offers the possibility to exchange buffers while maintaining beads and cells in their respective compartments. This feature allows to wash away contaminant extracellular mRNAs as well as performing all downstream enzymatic reactions in situ (RT and exonuclease treatments) facilitating the library preparation.

The overall design of the microfluidics chip is very innovative and the capability of the platform to preserve most of the cell input could be extremely useful for many researchers doing single cell sequencing of rare cell type. While the authors did an excellent work in demonstrating the exceptional cell loading performance and the overall controllability of the system, the part of the manuscript addressing the mRNA capture efficiency is notably less convincing. Prior to publication, I recommend to improve the analysis addressing the sensitivity of the system as well as few additional points:

Major

1) The analysis of Paired-seq sensitivity is inconsistent across the two analyses performed, and needs to be addressed to avoid confusion. In Figure 4E, using a standard ERCC analysis similar to Grun et al. (Nat Methods 2014), the authors argue they have 16.8% capture efficiency. However, in Figure 4G, the authors perform a different analysis using Svensson et al (Nat Methods 2017) approach and suggest the capture efficiency is closer to one in two molecules, or 50%. The data are inconsistent in the two panels—in Figure 4G, at a depth of 1 million reads per bead, the reported Paired-seq efficiency is far higher than the 16.8% calculated in Figure 4E. It is worth noting that a 50% mRNA sensitivity is extremely high, and will require substantial justification in order to be believed by the single cell community. Nowhere do the authors explicitly show the distribution of nUMIs/nGenes for a set of cells. If 50% is truly the efficiency, those distributions will be extremely high—for mES cells it would be expected that the median capture per cell would be in the tens of thousands (see PMID 29622030).

There are several claims in the paper that are rather hyperbolic. To describe a 16.8% capture efficiency as “much higher” than Drop-seq, when it is within a factor of 1.3, is excessive.

2) On line 298, the authors argue that the Pearson correlations are higher than Drop-seq “due to the active mixing and higher mRNA capture efficiency in picoliter.” However, this has never been actually shown—there are several differences in the Paired-seq protocol compared to the Drop-seq protocol. This sort of claim should be omitted unless it has been specifically experimentally tested. Noise reduction is crucial in single cell sequencing technology and a more careful examination of this question would be very valuable. It is possible that the ability to perform the enzymatic reactions in the chip’s chamber participates in the reduction of the technical noise. This hypothesis can be easily tested, by collecting the beads before reverse transcription and preparing the cDNA library according to the Drop-seq protocol.

3) A key feature in single cell sequencing is the throughput and it is not obvious whether this

system is easily scalable. Could authors demonstrate construction of a library in the tens of thousands of cells in size, or at least state clearly the throughput limitations?

4) Figure 4A is a bit confusing; the authors state that they only harvested 800 cell barcodes, meaning that the sum total of barcodes in the sequencing run should be 800 (there are no empty barcodes). Is it possible that most of the barcodes beyond the $x=768$ inflection point are simply mutations (from PCR, synthesis, sequencing)? I think this is supported by the fact that most of the points after the inflection point are actually high specificity (if I am reading the colors correctly). Since you have such a clear, precise estimate of the number of barcodes present, it should be possible to do far better in matching up barcodes than simply to use an inflection point in the cumulative distribution plot.

5) Will the authors be providing chip fabrication protocols, designs, and running protocols online? This method could be very useful to a lot of researchers, and hence the impact of their work would be significantly higher if they provide an open-source ability to adopt in other labs.

6) The system relies on change in flow resistance induces by cells when they load into the chambers, raising the question whether the system is sensitive to cell size. Nowadays, Nuc-seq is used by many researchers working with frozen tissues, and if the authors could show that the loading and the gene capture works equally well with nuclei, it would make this method more impactful.

7) The title is misleading because the cells and the beads are not encapsulated into droplets per se, rather in picoliter chambers.

Minor

a) Line 249 authors state they used 1 million of ERCC copy per chamber but on line 469 it is written 100K molecules per chamber.

b) Typo line 315 (should be figure 4I/J)

c) line 261. Marker gene with (low?) expression level

Reviewer #2:

Remarks to the Author:

Indeed, because of cellular compartmentalization (into droplets or wells) according to Poisson statistics, there is a long-standing problem in the field of high-throughput Single Cell (SC) genomics where profiling a precious few cells are indeed challenging. As the authors point out, CTCs in particular have been challenging to high-throughput platforms like Drop-seq, InDrop, SeqWell, etc. The work presented here has several merits and the potential to overcome some long-standing challenges in the Single cell RNA-seq.

However, there are several problems that the current manuscript fails to address that will prevent it from being put to use in the context of a few precious cells where profiling each one is important and losing even a single one is not an option. I will try to list them here:

Hardly ever do profiling large number of cells (throughput) and doing so for each and every one of them go hand in hand. When throughput is required, # cells needed for the experiment is hardly a consideration, e.g., PBMC, tumor microenvironment, etc., vs., CTCs, where every cell counts. When the number of cells is limited, 96-well plate based assays (manual or with liquid handling) are viable options.

The differential flow resistance principle of loading cells and beads in capture ports proposed by the authors is not new; the Fluidigm C1 SC RNA-seq platform which was one of the first

commercial platforms to offer single cell RNA-seq solution has been using this principle since 2012. Not only do the authors not acknowledge this, but they also do not acknowledge the problem associated with such capture chambers, which has severely restricted its use. As was seen in Fluidigm C1 system, the cells' dimensions play an important role in cell capture efficiency. In such designs, the cell capture ports are sensitive to cell size, being optimal for cells of a particular size. This makes it extremely difficult for users to determine which chip to use, even when microfluidic chips with multiple such dimensions are available. (Eventually, different chips targeting cells in three size ranges: 5-10, 10-17, and 17-25 microns, were launched. Incidentally, for applications where the number of cells available is limited, e.g., CTCs, the Fluidigm C1 is still a very viable option.)

As current standard, the SC RNA-seq field has moved to profiling single cells from complex tissues where there are cells in a range of sizes. For such purposes, this method becomes tricky because there is no 'average' cell size to target. The onus of demonstration is indeed higher than it used to be 2-4 years back as the field matured and the SC techniques have been put to more and more sophisticated uses. The most common technique now is 10x genomics Chromium. It would be useful to compare against this technique where the problem of cell-size does not exist. Very little information on the microfluidic chip has been provided in terms of dimensions, operation time, loading time, etc. In terms of novelty of SC findings, the work is not novel and does not reveal anything that was not expected. If anything, the value of the manuscript would be in demonstrating the usefulness of the technique as a resource for the community which the paper fails to do, due to lack of information and experimental details that will prevent its readers from implementing the technique in their labs.

Claims of throughput have not been demonstrated. While theoretically possible (authors do not provide the number of capture ports in the devices used or their dimensions; valuable information that has been omitted and is frustrating to the reader) there needs to be a demonstration and discussion on how many chips were run to generate each dataset, how many experiments were pooled to generate the data, etc. Again, because the needs of the SC field have matured beyond simple demonstrations on cell lines to processing fragile tissues where cell viability is limited and decreases with handling time, it is up to the authors to provide such demonstration to establish utility of their technique. It certainly does not seem easy to fabricate or run this microfluidic device. The returns have to be significant for the pains to be worth it.

In spite of the above drawbacks, there are some advantages that the 'Paired Seq' technique offers that are worthy of mention.

The microfluidic chip itself is sophisticated. The low amount of crosstalk between cells is a very attractive feature.

The low sensitivity claim here is astonishing. This in itself would make the technique very valuable. One of the major drawbacks of current SC RNA-seq techniques is that they are terrible at detecting rare transcripts. If the 2-molecule sensitivity is indeed true as claimed, the value of this technique would be in detecting low abundance transcripts and improving drop-out rates. Analysis of genes that are expressed at low levels would be very informative, as would be comparison with the sensitivity of spatial transcriptomics techniques which are better suited for this detection of lowly expressed genes.

I suspect the high # genes captured at low read depth (if confirmed) is because the authors wash away the ambient RNA in solution in the cell suspension loaded. This could use further investigation, for example how jackpotting events are avoided here. Does exonuclease I reaction performed on chip help? Have the authors done comparisons of capture efficiency and read depth for exo reactions performed on and off chip?

The manuscript needs more experimental details in general. Dimensions of cell and bead chambers, etc., time taken to load and run experiment, how the chip is heated, whether the chip has internal heating element, etc. would be helpful. Better description of chip fabrication and operation is requested.

What I would like to see to consider the manuscript of broad utility and thus suitable for publication:

- 1) Demonstrate the technique on primary cells of different size with least 2000-5000 cells per run (not unreasonable because Fluidigm C1 mRNA-seq HT IFC <https://www.fluidigm.com/products/c1->

system processes 800 cells at a time which is a comparable platform to Paired-Seq).

2) Establish that the long loading time at 0.02 mL/hr, as must be required to prevent high shear forces from damaging cells, is not detrimental to RNA quality or spurious/stress related transcription.

Also note: Statistical analyses used here are sufficiently convincing. The comparison between different techniques is also valuable and putting the current technique in context is valuable.

Response to Reviewers

Reviewer 1:

Comments:

Zhang et al. and colleagues describe a novel microfluidics platform enabling single cell sequencing with very minimal cell loss. The crux of the approach lies in efficient control of cell and bead loading into individual chambers using differential flow resistance. This new method results in a near perfect co-encapsulation ratio 1:1 (beads:cells) while maintaining the doublet rate at a very low level (<1%). To achieve a comparable doublet rate, a typical Poisson loading would result in ~90% of cell and bead loss.

Beside preserving most of the cell input, this system offers the possibility to exchange buffers while maintaining beads and cells in their respective compartments. This feature allows to wash away contaminant extracellular mRNAs as well as performing all downstream enzymatic reactions in situ (RT and exonuclease treatments) facilitating the library preparation.

The overall design of the microfluidics chip is very innovative and the capability of the platform to preserve most of the cell input could be extremely useful for many researchers doing single cell sequencing of rare cell type. While the authors did an excellent work in demonstrating the exceptional cell loading performance and the overall controllability of the system, the part of the manuscript addressing the mRNA capture efficiency is notably less convincing. Prior to publication, I recommend to improve the analysis addressing the sensitivity of the system as well as few additional points:

Author's response: We thank the reviewer's careful reading and positive remarks on the novelty of the work. We have carefully considered reviewers' comments and revised the manuscript accordingly.

Major point:

1. The analysis of Paired-seq sensitivity is inconsistent across the two analyses performed, and needs to be addressed to avoid confusion. In Figure 4E, using a standard ERCC analysis similar to Grun et al. (Nat Methods 2014), the authors argue they have 16.8% capture efficiency. However, in Figure 4G, the authors perform a different analysis using Svensson et al (Nat Methods 2017) approach and suggest the capture efficiency is closer to one in two molecules, or 50%. The data are inconsistent in the two panels—in Figure 4G, at a depth of 1 million reads per bead, the reported Paired-seq efficiency is far higher than the 16.8% calculated in Figure 4E.

It is worth noting that a 50% mRNA sensitivity is extremely high, and will require substantial justification in order to be believed by the single cell community. Nowhere do the authors explicitly show the distribution of nUMIs/nGenes for a set of cells. If 50% is truly the efficiency, those distributions will be extremely high—for mES cells it would be expected that the median capture per cell would be in the tens of thousands (see PMID 29622030).

There are several claims in the paper that are rather hyperbolic. To describe a 16.8% capture efficiency as “much higher” than Drop-seq, when it is within a factor of 1.3, is excessive.

Authors' response: We thank the reviewer for pointing this out. As the reviewer mentioned, there are two different data analysis methods used in our manuscript (Cell 2015 and Nat. Methods 2014/2017). The capture efficiency differences showed in Figure 4E and 4G are due to UMI data processing differences of these two analysis methods. The difference between these two methods lies in the merging of single-base difference UMIs (performed in Cell 2015 (Drop-seq) but not in Nat Methods 2014 or Nat Methods 2017). We performed UMI merging in Figure 4E but not in Figure 4G. According to the statistical results, the probability of only one base difference between different UMIs is very low. UMIs merging will be more reasonable, so the capture efficiency of Paired-seq described in the manuscript was 16.8% (Figure 4E). Unfortunately, such a dispose of UMI merging is not applicable for sequencing platforms without using UMIs such as Smart-seq. To ensure a fair comparison of different sequencing platforms with UMIs (such as Drop-seq) and without UMIs, we followed the same protocol as reported by Svensson et al (Nat Methods 2017) that processing data without UMI merging and yielded a capture efficiency of 50% (2 molecules) as shown in Figure 4G.

Although data look different, these two values are reasonable and comparable to those obtained for Drop-seq. For Drop-seq data, it has also been reported in Cell 2015 to have two different capture efficiencies using these two different analysis methods. A capture efficiency of about 12.8% was reported with UMI merging, while an efficiency of 47% was reported without UMI merging. In comparison, capture efficiencies we obtained for Paired-Seq were 16.8% and 50% respectively, which are comparable and slightly higher than Drop-seq.

To avoid confusion, we have added following discussion in the revision: “Sequencing data from Paired-seq were analyzed with two established data analysis methods, with UMI merging and without merging, yielding capture efficiencies of 16.8% and 50% respectively. Both values are slightly higher than those of Drop-seq (12.8% and 47%) respectively”.

We have also changed “much higher” to “slightly higher” in the revision.

2. On line 298, the authors argue that the Pearson correlations are higher than Drop-seq “due to the active mixing and higher mRNA capture efficiency in picoliter.” However, this has never been actually shown—there are several differences in the Paired-seq protocol compared to the Drop-seq protocol. This sort of claim should be omitted unless it has been specifically experimentally tested. Noise reduction is crucial in single cell sequencing technology and a more careful examination of this question would be very valuable. It is possible that the ability to perform the enzymatic reactions in the chip’s chamber participates in the reduction of the technical noise. This hypothesis can be easily tested, by collecting the beads before reverse transcription and preparing the cDNA library according to the Drop-seq protocol.

Author's response: We thank the reviewer's careful reading and kind suggestions. We have omitted this claim.

Indeed, the ability to perform the enzymatic reactions on chip participates in the reduction of the technical noise. We compared the quality of sequencing data for ERCC experiment by using Paired-seq with enzymatic process on and off chip (in tube). It was found that percentages of mapped reads for enzymatic process on chip were significantly higher than that in tube, which

confirmed that the enzymatic operation on chip could remove technical noise. This result has been added as Supplementary Figure 11 in the revision.

3. A key feature in single cell sequencing is the throughput and it is not obvious whether this system is easily scalable. Could authors demonstrate construction of a library in the tens of thousands of cells in size, or at least state clearly the throughput limitations?

Author's response: We thank the reviewer's kind suggestion. The chip can be easily scaled up. We have designed and fabricated a Paired-seq chip with 2000 paired units. The results have been added as Supplementary Figure 2 in the revision.

4. Figure 4A is a bit confusing; the authors state that they only harvested 800 cell barcodes, meaning that the sum total of barcodes in the sequencing run should be 800 (there are no empty barcodes). Is it possible that most of the barcodes beyond the $x=768$ inflection point are simply mutations (from PCR, synthesis, sequencing)? I think this is supported by the fact that most of the points after the inflection point are actually high specificity (if I am reading the colors correctly). Since you have such a clear, precise estimate of the number of barcodes present, it should be possible to do far better in matching up barcodes than simply to use an inflection point in the cumulative distribution plot.

Author's response: We thank the reviewer's kind comments. Actually we used 800-units chip instead of harvesting 800 cell barcodes in the article (Line 228-229). Inflection point is a standard method to estimate the cell barcodes, which is commonly used in scRNA-seq such as 10x, Seq-well, Drop-seq, inDrop. Although we could parallel process single cell/single bead pairing with very high efficiency, due to the errors coming from subsequent amplification and sequencing process, we found that the result of the sequencing still generated a large number of cell barcodes. As the reviewer mentioned, the barcodes beyond the $x=768$ inflection point are actually simply mutations (from PCR, synthesis, or sequencing), since the number of UMIs for each cell barcode beyond inflection point was only dozens to hundreds, far less than the amount of mRNAs of a single cell. Furthermore, the high specificity after the inflection point also demonstrated the complete removal of background cell-free mRNAs in the cell suspension on Paired-seq chip because there was low probability for cell sources of different species to mutate into the same cell barcode sequence.

5. Will the authors be providing chip fabrication protocols, designs, and running protocols online? This method could be very useful to a lot of researchers, and hence the impact of their work would be significantly higher if they provide an open-source ability to adopt in other labs.

Author's response: We thank the reviewer's kind suggestion. In the revision, we have provided a more detailed chip fabrication, design and running protocols in the Experimental section and also in supplementary materials (Supplementary Figure 3).

6. The system relies on change in flow resistance induces by cells when they load into the chambers, raising the question whether the system is sensitive to cell size. Nowadays, Nuc-seq is

used by many researchers working with frozen tissues, and if the authors could show that the loading and the gene capture works equally well with nuclei, it would make this method more impactful.

Author's response: We thank the reviewer's kind suggestion. The system is not sensitive to cell size. Using a chip with 5 μm \times 5 μm orifice of the capture chamber, cells with different average sizes ranging from 10, 15, to 25 μm were able to be captured with a similar capture efficiency, establishing that size problem does not exist for Paired-seq. The results have been added in the revision as Supplementary Figure 4. Theoretically, any particles larger than the orifice of capture chamber can be trapped and analyzed. In this regard, our chip can be adapted for Nuc-seq as long as the orifice can be fabricated to submicron level. However, this will require very expensive nano-fabrication facility, which is beyond the scope of this manuscript.

7. The title is misleading because the cells and the beads are not encapsulated into droplets per se, rather in picoliter chambers.

Author's response: We thank the reviewer's kind suggestion. In the revision, the title has been changed to "Highly Parallel and Efficient Single Cell mRNA Sequencing with Paired Picoliter Chambers."

Minor

a) Line 249 authors state they used 1 million of ERCC copy per chamber but on line 469 it is written 100K molecules per chamber.

Author's response: We thank the reviewer for pointing out this typo. We thank the reviewer for pointing out this typo. We have fixed this by changing "1 million copies" to "About 100K molecules" in the revision.

b) Typo line 315 (should be figure 4I/J)

Author's response: We thank the reviewer for pointing out this typo. We have changed the typo line 315 "Figure 5I/J" to "Figure 4I/J" in the revision.

c) line 261. Marker gene with (low?) expression level

Author's response: We thank the reviewer's kind suggestion. We have changed the "maker gene with expression level" to "marker genes with low expression level."

Reviewer 2:

Comments:

Indeed, because of cellular compartmentalization (into droplets or wells) according to Poisson statistics, there is a long-standing problem in the field of high-throughput Single Cell (SC) genomics where profiling a precious few cells are indeed challenging. As the authors point out, CTCs in particular have been challenging to high-throughput platforms like Drop-seq, InDrop, SeqWell, etc. The work presented here has several merits and the potential to overcome some long-standing challenges in the Single cell RNA-seq.

Author's response: We thank the reviewer's careful reading and positive remarks on the novelty of the work. We have carefully considered the comments and revised the manuscript accordingly.

However, there are several problems that the current manuscript fails to address that will prevent it from being put to use in the context of a few precious cells where profiling each one is important and losing even a single one is not an option. I will try to list them here:

Hardly ever do profiling large number of cells (throughput) and doing so for each and every one of them go hand in hand. When throughput is required, # cells needed for the experiment is hardly a consideration, e.g., PBMC, tumor microenvironment, etc., vs., CTCs, where every cell counts. When the number of cells is limited, 96-well plate based assays (manual or with liquid handling) are viable options.

Author's response: We thank the reviewer's kind comments. Our current design can process 800 cells in one chip. This can be easily scaled up to thousands of cells per chip as shown in Supplementary Figure 2. As a result, one can easily process thousands of single cells using one chip or several chips in parallel.

We agree with the reviewer that when the number of cells is limited, 96-well plate based assay is an option. However, 96-well plate based assay requires manual operation, large reaction volume, resulting in higher reagent consumption, higher contamination possibility, lower sensitivity, and more expensive. In contrast, the reaction chamber of Paired-seq chip is in picoliter range, thus resulting in lower reagent consumption, lower contamination probability, higher sensitivity, and more cost effective. More importantly, for CTC analysis, the number of cells isolated from a patient in many cases could go higher than 96 due to the imperfect isolation efficiency of current enrichment platforms which results in contamination of large number of white blood cells. A platform capable of analyzing hundreds of cells is thus more compatible to CTC analysis.

The differential flow resistance principle of loading cells and beads in capture ports proposed by the authors is not new; the Fluidigm C1 SC RNA-seq platform which was one of the first commercial platforms to offer single cell RNA-seq solution has been using this principle since 2012. Not only do the authors not acknowledge this, but they also do not acknowledge the problem associated with such capture chambers, which has severely restricted its use. As was seen in Fluidigm C1 system, the cells' dimensions play an important role in cell capture efficiency. In such designs, the cell capture ports are sensitive to cell size, being optimal for cells of a particular size. This makes it extremely difficult for users to determine which chip to use, even when microfluidic chips with multiple such dimensions are available. (Eventually, different chips targeting cells in three size ranges: 5-10, 10-17, and 17-25 microns, were launched. Incidentally, for applications where the number of cells available is limited, e.g., CTCs, the Fluidigm C1 is still a very viable option.) As current standard, the SC RNA-seq field has moved to profiling single cells from complex tissues where there are cells in a range of sizes. For such purposes, this method becomes tricky because there is no 'average' cell size to target. The onus of demonstration is indeed higher than it used to be 2-4 years back as the field matured and the SC techniques have been put to more and more sophisticated uses. The most common technique now is 10x genomics

Chromium. It would be useful to compare against this technique where the problem of cell-size does not exist.

Author's response: We thank the reviewer for pointing this out. The differential flow resistance principle of loading particles such as cells and beads in capture ports indeed had been reported as we cited in our manuscript (ref. 31). However, principle of pairing single cell and single bead reported in this manuscript has never been reported. The innovation of our platform design lies in the realization of efficient capture and pairing of cell and barcoded bead. Although both Paired-seq and Fluidigm C1 system use hydrodynamic differential flow resistance to capture cell, Paired-seq is unique in the pairing mode of cell and bead. Compared to our design, Fluidigm C1 cannot pair one cell with one bead, thus it requires the use of liquid barcodes, which is expensive to synthesize, tedious to handle and limited in throughput.

For particle trapping, two systems use different channel designs. Fluidigm C1 use valves to isolate each reaction chamber, which required fixed channel height for optimal isolation of one reaction chamber from the other. Moreover, cell size is more sensitive to the change of flow resistance due to the close distance between the capture site and the site of differential flow rate change in the chip design of Fluidigm C1. As a result, Fluidigm C1 required different chips targeting cells in three size ranges (5-10, 10-17 and 17-25 μm), unfeasible to simultaneously analyze different sizes of cells at a time. In contrast, the capture of cells and barcoded bead is completely independent on Paired-seq chip. Cell larger than 5 μm can be processed on Paired-seq chip because the cell trap was designed and fabricated with 5 $\mu\text{m} \times 5 \mu\text{m}$ (width and height) by multiple-step lithography. And we have tested cells with different average sizes changing from 5 μm to 40 μm , including small size of cell line (3T3 cells, 5-25 μm), medium size of cell line (K562, 10-26 μm), and large size of cells (drug (*Nocodazole*) treated K562 cells, 15-40 μm). We did not see any differences of capture efficiency among them. These results established that size problem does not exist for Paired-seq. The results have been added in the revision as Supplementary Figure 4.

Very little information on the microfluidic chip has been provided in terms of dimensions, operation time, loading time, etc. In terms of novelty of SC findings, the work is not novel and does not reveal anything that was not expected. If anything, the value of the manuscript would be in demonstrating the usefulness of the technique as a resource for the community which the paper fails to do, due to lack of information and experimental details that will prevent its readers from implementing the technique in their labs.

Author's response: We thank the reviewer's kind suggestion. We have provided a detailed and easy to followed protocol on chip fabrication, design and operation in the experimental section and supplementary materials (Supplementary Figure 3).

Claims of throughput have not been demonstrated. While theoretically possible (authors do not provide the number of capture ports in the devices used or their dimensions; valuable information that has been omitted and is frustrating to the reader) there needs to be a demonstration and discussion on how many chips were run to generate each dataset, how many experiments were pooled to generate the data, etc. Again, because the needs of the SC field have matured beyond

simple demonstrations on cell lines to processing fragile tissues where cell viability is limited and decreases with handling time, it is up to the authors to provide such demonstration to establish utility of their technique. It certainly does not seem easy to fabricate or run this microfluidic device. The returns have to be significant for the pains to be worth it.

Author's response: We thank the reviewer's kind suggestion. The chip can be easily scaled up. We have designed and fabricated a Paired-seq chip with 2000 paired units. The result has been added as Supplementary Figure 2 in the revision. And we have provided a detailed and easily followed protocol on chip design, fabrication and operation in the experimental section and supplementary materials (Supplementary Figure 1-3 and Supplementary Movie 0). For all the sequencing results, each dataset was generated from one single chip. One single experiment was pooled to generated one data. We have added the details in the revision. This work is for demonstrating the value of the technique as a resource for the community. This new method demonstrates high cells/beads utilization efficiency, cell-free RNAs removal capability, high gene detection ability and low cost, which will have broad applications for scRNA-seq.

In spite of the above drawbacks, there are some advantages that the 'Paired Seq' technique offers that are worthy of mention.

The microfluidic chip itself is sophisticated. The low amount of crosstalk between cells is a very the attractive feature.

The low sensitivity claim here is astonishing. This in itself would make the technique very valuable. One of the major drawbacks of current SC RNA-seq techniques is that they are terrible at detecting rare transcripts. If the 2-molecule sensitivity is indeed true as claimed, the value of this technique would be in detecting low abundance transcripts and improving drop-out rates. Analysis of genes that are expressed at low levels would be very informative, as would be comparison with the sensitivity of spatial transcriptomics techniques which are better suited for this detection of lowly expressed genes.

I suspect the high # genes captured at low read depth (if confirmed) is because the authors wash away the ambient RNA in solution in the cell suspension loaded. This could use further investigation, for example how jackpotting events are avoided here. Does exonuclease I reaction performed on chip help? Have the authors done comparisons of capture efficiency and read depth for exo reactions performed on and off chip?

Author's response: We thank the reviewer's positive remarks on the novelty and advantages of the Paired-seq technique. We also had compared the quality of sequencing data for ERCC experiment by using Paired-seq with enzymatic process on and off chip (in tube). It was found that the percentages of mapped reads for enzymatic process on chip was significantly higher than those in the tube, which confirmed that the enzymatic operations on chip could remove technical noise. The result was added as Supplementary Figure 11 in the revision.

The manuscript needs more experimental details in general. Dimensions of cell and bead chambers, etc., time taken to load and run experiment, how the chip is heated, whether the chip has internal heating element, etc. would be helpful. Better description of chip fabrication and operation is requested.

Author's response: We thank the reviewer's kind suggestion. We have provided more experimental details including chip fabrication, design and operation protocols in the experimental section and supplementary materials (Supplementary Figure 1-3 and Supplementary Movie 0).

What I would like to see to consider the manuscript of broad utility and thus suitable for publication:

1. *Demonstrate the technique on primary cells of different size with least 2000-5000 cells per run (not unreasonable because Fluidigm C1 mRNA-seq HT IFC <https://www.fluidigm.com/products/c1-system> processes 800 cells at a time which is a comparable platform to Paired-Seq).*

Author's response: We thank the reviewer's kind suggestion. The chip can be easily scaled up. We have designed and fabricated a Paired-seq chip with 2000 paired units. The result has been added as Supplementary Figure 2 in the revision.

2. *Establish that the long loading time at 0.02 mL/hr, as must be required to prevent high shear forces from damaging cells, is not detrimental to RNA quality or spurious/stress related transcription.*

Author's response: We thank the reviewer's kind suggestion. To test the influence of shear forces on RNA quality or transcription, we compared the gene detection ability reflecting RNA integrity with different loading time. Our results showed that at the loading time of 15 min and 40 min, the number of detected genes show no significant difference. This data suggest that long loading time was not detrimental to RNA quality or spurious/stress related transcription.

Also note: Statistical analyses used here are sufficiently convincing. The comparison between different techniques is also valuable and putting the current technique in context is valuable.

Author's response: We thank the reviewer's positive remarks.

Reviewers' Comments:

Reviewer #1:

Remarks to the Author:

The revised manuscript continues to show concerning mismatches between the data shown, and the conclusions drawn by the authors about that data. First and foremost, the capture efficiency continues to be confusing and potentially misleading to readers. Based on the nUMI distributions for the mES cells shown in Supplemental Figure 16, there is absolutely no way that the absolute capture efficiency of this technology is 50%, or "1 in 2 molecules" as described by the authors. From published work by Klein et al., 2015 (PMID 26000487) who sampled the same ES cells as were used by the current authors, they clearly state: "The average number of reads per cell ranged up to 208×10^3 , and the average UMIFM count up to 29×10^3 ." Klein et al computed a capture efficiency of their technology—inDrop—as less than 10%. How can the authors, whose maximum number of UMIs measured in a single cell is less than Klein et al's reported average, credibly argue that their efficiency is 1 in 2?

Instead of such claims, I would instead suggest that the authors simply show their analysis, and state that the efficiency of Paired-seq is comparable to other modern single-cell approaches.

Supplemental Figure 2, added in response to my comment (and similar to Reviewer #2's comment) about capture efficiency, is rather insufficient, showing only a set of four contrast images showing different zones of the 2000-unit chip. Does this chip actually operate? If Paired-seq is as inexpensive and easy as the authors contend, could they simply perform one human-mouse experiment with this chip to demonstrate its functionality? Sequencing could be quite shallow, just enough to show human-mouse species specificity—1 million reads would really be sufficient (I am not trying to demand expensive experiments here). Can the authors clearly explain what the tradeoffs would be to, for example, generating a 10,000 unit chip to be competitive with the throughput of the 10X Chromium system?

The authors show that the chip is capable of capturing cells ranging in size from 10 microns to 25 microns, from which they state that "the system is not sensitive to cell size." However, many, if not most, primary cells from tissues are smaller than 10 microns—for example mammalian PBMCs are all smaller than 10 microns. The authors suggest that nuclei are not feasible in the current system because they are too small and would require a special fabrication approach to generate, so there are some size limitations. Could the authors clearly state what the functioning size range is for their system, and how this might impact how a reader would choose to try Paired-seq in their particularly experimental/biological system?

Reviewer 2 had a nice suggestion to add an exonuclease I reaction on-chip and compare capture efficiency and read depth of these data. However, the authors replied with a figure describing differences in mapping rates, which didn't really address the reviewer's suggestion. It is also very confusing why, molecularly, an on-chip versus off-chip Exonuclease I reaction would affect mapping rate.

Less important, but nonetheless necessary to revise:

Line 16, "...because of the Poisson distribution": The distribution itself is not responsible for the low capture efficiency of droplet-based methods. Poisson statistics are used across a wide range of λ . It's the need for limiting dilution into droplets that's the problem. The authors need to clarify this here and elsewhere (e.x. Line 20).

Line 253: "Paired-seq" presented massively parallel and high-efficient single-cell mRNA sequencing..." The throughput demonstrated here is not "massively parallel" by modern standards. Numerous single-cell papers currently published in 2018 and 2019 include hundreds of thousands and sometimes millions of cell profiles. This is another example of hyperbolic claims that can

mislead readers.

Other:

Typo line 230 "Results suggestion"

Reviewer #2:

Remarks to the Author:

I thank the authors for revising the manuscript as per referee comments. However, there are several outstanding issues that the authors have failed to address satisfactorily in the revised manuscript that prevents me from recommending the manuscript for publication in Nature Communication.

One of the most attractive claims of the manuscript previously (to me) that it has high sensitivity for low abundance transcripts has failed to hold. Upon clarification to Referee 1's comments, we find that the sensitivity for the technique is only slightly higher than Drop-seq "yielding capture efficiencies of 16.8% and 50% respectively... slightly higher than those of Drop-seq (12.8% and 47%) respectively" (Drop-seq is not that sensitive in itself; the commercially available Chromium platform from 10x Genomics does much better, albeit at higher cost). In light of this new clarification, the sensitivity of "~2 molecules" may also need to be revisited.

Again, the needs of the single cell genomics field have matured beyond simple demonstrations on cell lines to processing fragile tissues where cell viability is limited and decreases with handling time. It is simply not enough to demonstrate the 'paired-seq' technique on cell lines where cells are hardy and well behaved. The authors claim that loading time of 15 min and 40 mins (indicated in referee response but not mentioned in manuscript) does not affect the number of detected genes in k562 and 3T3 cells. But this can be significantly detrimental to viability for cells harvested from primary tissues, that will be seen not directly from # genes detected but from higher expression of ribosomal and stress response genes. I am also not convinced that a 'one size fits all' chip design will work for cells from primary tissues where different cell types come in different cell-sizes.

The manuscript still does not have enough experimental details on how the chip is operated. What kind of pumps, valves and equipment were used for the gas phase flow? How was the chip incubated? What volumes of enzymes were needed to load the 800-cell and 2000-cell chip? What are the dead volumes? CAD files for the chip and schema for the equipment would be useful to understand the setup and workflow.

'Paired-seq' is a sophisticated technique. The manuscript is otherwise well-written and the authors have made substantial effort in comparing their technique to other single cell techniques. I am not convinced that the manuscript as it stands is impactful and of broad interest to the audience of Nature Communications, but it may be well suited for publication in Nature Biotechnology or Nature Genetics where practitioners of the single-cell genomics field may find the technique useful.

Reviewers' comments:

Reviewer #1 (Remarks to the Author):

1. *The revised manuscript continues to show concerning mismatches between the data shown, and the conclusions drawn by the authors about that data. First and foremost, the capture efficiency continues to be confusing and potentially misleading to readers. Based on the nUMI distributions for the mES cells shown in Supplemental Figure 16, there is absolutely no way that the absolute capture efficiency of this technology is 50%, or “1 in 2 molecules” as described by the authors. From published work by Klein et al., 2015 (PMID 26000487) who sampled the same ES cells as were used by the current authors, they clearly state: “The average number of reads per cell ranged up to 208×10^3 , and the average UMIFM count up to 29×10^3 .” Klein et al computed a capture efficiency of their technology—inDrop—as less than 10%. How can the authors, whose maximum number of UMIs measured in a single cell is less than Klein et al’s reported average, credibly argue that their efficiency is 1 in 2?*

Instead of such claims, I would instead suggest that the authors simply show their analysis, and state that the efficiency of Paired-seq is comparable to other modern single-cell approaches.

Author’s Response: We thank the reviewer’s careful reading and kind suggestions. We performed comparison of depth-associated sensitivity by using the method of Svensson et al. between Paired-seq and other platforms (Materials and methods). This method takes advantage of the big amount of data from 15 scRNA-seq platforms together to model the relationship between detection limit and sequencing depth of scRNA-seq platforms to make a fair evaluation when the data of each platform on each sequencing depth are insufficient. The basic hypothesis of this method is that the detection limit of each scRNA-seq platform decreases as the increasing of sequencing depth, and reaches a plateau at a constant sequencing depth. This method provided relatively stable and uniform metrics for the comparison of various scRNA-seq platforms, and achieved the detection limit of such as "10 molecules" for Drop-seq, "5 molecules" for inDrop, and also "2 molecules" for Paired-seq. However, this algorithm could not provide an absolute value of sensitivity but a relative value just for comparison, which brings confusion to reviewers. To remove this confusion, we have deleted the claim of 2 molecules detection limit and changed the expression to “The algorithm provided a fair comparison and demonstrated that sensitivity of Paired-seq was comparable to other modern single-cell approaches.” in the revision.

2. *Supplemental Figure 2, added in response to my comment (and similar to Reviewer #2's comment) about capture efficiency, is rather insufficient, showing only a set of four contrast images showing different zones of the 2000-unit chip. Does this chip actually operate? If Paired-seq is as inexpensive and easy as the authors contend, could they simply perform one human-mouse experiment with this chip to demonstrate its functionality? Sequencing could be quite shallow, just enough to show human-mouse species specificity—1 million reads would really be sufficient (I am not trying to demand expensive experiments here). Can the authors clearly explain what the tradeoffs would be to, for example, generating a 10,000 unit chip to be competitive with the throughput of the 10X Chromium system?*

Author's Response: We thank the reviewer's careful reading and kind suggestions. We performed a human- mouse mix experiment on the 2000-unit Paired-seq chip, and provided the sequencing results in the Supplementary Figure S2E, which demonstrated the feasibility of the 2000-unit Paired-seq chip. For all the hydrodynamic based microfluidic chips for single particle capture, the area of patterned channel on chip is the key factor for throughput, and it is difficult to guarantee the uniformity of the pattern for large area due to the inherent defects and limitations of lithography technology for the fabrication of SU-8 template, such as the area, perpendicularity and uniformity of UV-irradiation. So, it is difficult to generate 10,000-unit chip competitive with the throughput of the 10X Chromium system. The advantages of Paired-seq chip are mainly reflected in the efficient pairing ability, low reagent consumption, high detection sensitivity and accuracy, as well as the background removal ability and in situ observability, etc. Therefore, we aim to expand the application of Paired-seq chip instead of increasing the throughput, such as multi-omics analysis.

3. *The authors show that the chip is capable of capturing cells ranging in size from 10 microns to 25 microns, from which they state that "the system is not sensitive to cell size." However, many, if not most, primary cells from tissues are smaller than 10 microns—for example mammalian PBMCs are all smaller than 10 microns. The authors suggest that nuclei are not feasible in the current system because they are too small and would require a special fabrication approach to generate, so there are some size limitations. Could the authors clearly state what the functioning size range is for their system, and how this might impact how a reader would choose to try*

Paired-seq in their particularly experimental/biological system?

Author's Response: We thank the reviewer's careful reading and kind suggestions. What we claimed in the manuscript and previous responses to the reviewers was that the chip was capable of capturing cells ranging in size from 5 to 40 μm that was suitable to most of the mammalian cell capture such as PBMC. In order to confirm the capture capability, we measured the diameter of captured cells in Paired-seq chip shown in Supplementary Figure 2A. The cell diameter was measured as follows: 24.5 μm , 18.6 μm , 13.2 μm , 14.8 μm , 23.4 μm from left to right for the upper cell capture channel and 8.6 μm , 15.0 μm , 16.4 μm , 5.5 μm , 20.8 μm from left to right for the lower cell capture channel.

What is more, the reviewer gave us a nice suggestion for conducting Nuc-seq on Paired-seq chip in the previous review. We performed a nucleus and cell line capture experiment. The mixture of nuclei (extracted by ExKineTM Nuclei Extraction Kit and stained with HOECHST) and SW480 cell line (stained with HOECHST for nucleus and DIL dye for cell membrane) was injected into the chip for single cells/ nuclei capture. The result showed that with the width of 3.58 μm gap, SW480 cell line (**Figure 1B**) and nuclei (**Figure 1C**) could be simultaneously captured by Paired-seq chip, demonstrating the ability to capture different sizes of cells/nuclei on a single chip and the feasibility to conduct Nuc-seq on Paired-seq chip. Thank the reviewer's kind suggestions to provide a new idea for the application of Paired-seq chip platform.

According to the results described above, we have fully demonstrated that we can capture and analyze cells of various sizes on a single chip, and the cell size ranges from 5 to 40 μm .

Figure 1. Characterization of chip size (A) and cell/ nuclei capture (B and C). Nuclei of SW480 cells were extracted by Nuclei Extraction Kit (ExKine™ Nuclei Extraction Kit, KTP4002). Cell nuclei were stained with HOECHST (blue), and cell membrane were stained with DIL dye (red).

4. *Reviewer 2 had a nice suggestion to add an exonuclease I reaction on-chip and compare capture efficiency and read depth of these data. However, the authors replied with a figure describing differences in mapping rates, which didn't really address the reviewer's suggestion. It is also very confusing why, molecularly, an on-chip versus off-chip Exonuclease I reaction would affect mapping rate.*

Author's Response: We thank the reviewer's careful reading and kind suggestions. Compared with mRNA (10^5 - 10^6 per cell) in a single cell, there are more than 10^8 primers on a single barcode bead. The number of primer is far more than mRNA in a single cell. After mRNA transcription, there are plenty of unextended primers left on the barcoded beads, which will take part in the following PCR reaction, resulting in a large amount of primer dimer (short reads). The Exonuclease I has the ability to remove the unextended primers. The removal of short sequences in solution can increase the proportion of truly effective reads (mapped reads), thus reducing the cost of sequencing. For bead based sequencing platform, in order to effectively perform nuclease digestion, barcoded beads need to be collected and placed on the rotary mixer for enzymatic digestion reaction, such as Drop-seq and Seq-well. However, because of the ability of in situ cleaning and enzymatic reaction of well-ordered barcoded beads for Paired-seq chip, direct Exonuclease I reaction on the chip can avoid the problems of insufficient reactants and incomplete enzymatic digestion caused by the accumulation of barcoded beads. More importantly, because of the picoliter reactor, which significantly increased local enzyme concentration thus digestion efficiency. To test the hypothesis, we had compared the quality of sequencing data for ERCC experiment by using Paired-seq with enzymatic process on and off chip (in tube). It was found that the percentages of mapped reads for enzymatic process on chip was significantly higher than those in the tube at the same sequencing depth (raw reads). Most of the unmapped reads (due to too short sequence) in off-chip sample could be traced back to primer on the barcoded beads, which confirmed that the insufficient enzymatic reaction brought in technical noise.

Additionally, we compared the capture efficiency of ERCC between on and off chip Exonuclease I reaction. The results showed that the capture efficiency for on chip Exonuclease I reaction (16.8%) was higher than off chip (15.3%). Therefore, we hold the opinion that the high gene detection capacity allows lower sequencing costs because it requires less sequencing depth to achieve the same number of detected genes described in the part of Discussion.

Less important, but nonetheless necessary to revise:

5. Line 16, “...because of the Poisson distribution”: *The distribution itself is not responsible for the low capture efficiency of droplet-based methods. Poisson statistics are used across a wide range of lambda. It’s the need for limiting dilution into droplets that’s the problem. The authors need to clarify this here and elsewhere (e.x. Line 20).*

Author’s Response: We thank the reviewer for pointing this out. We have changed the expression of “Poisson distribution” to “limited dilution for single-cell isolation” in Line 20.

6. Line 253: “Paired-seq” *presented massively parallel and high-efficient single-cell mRNA sequencing...* *The throughput demonstrated here is not “massively parallel” by modern standards. Numerous single-cell papers currently published in 2018 and 2019 include hundreds of thousands and sometimes millions of cell profiles. This is another example of hyperbolic claims that can mislead readers.*

Author’s Response: We thank the reviewer for pointing this out. We have deleted the expression of “massively parallel” in Line 253.

Other:

7. Typo line 230 “Results suggestion”

Author’s Response: We thank the reviewer for pointing this out. We have changed “Paired-seq shows” to “Results suggests that Paired-seq has the highest accuracy among all platforms with the predicted correlation $R = 0.955$ at the depth of 1 million reads per cell.”

Reviewer #2 (Remarks to the Author):

I thank the authors for revising the manuscript as per referee comments. However, there are several outstanding issues that the authors have failed to address satisfactorily in the revised manuscript that prevents me from recommending the manuscript for publication in Nature Communication.

Author Response: We thank the reviewers' valuable time evaluating our revision. We have carefully considered reviewers' comments and revised the last revision accordingly.

1. One of the most attractive claims of the manuscript previously (to me) that it has high sensitivity for low abundance transcripts has failed to hold. Upon clarification to Referee 1's comments, we find that the sensitivity for the technique is only slightly higher than Drop-seq "yielding capture efficiencies of 16.8% and 50% respectively... slightly higher than those of Drop-seq (12.8% and 47%) respectively" (Drop-seq is not that sensitive in itself; the commercially available Chromium platform from 10x Genomics does much better, albeit at higher cost). In light of this new clarification, the sensitivity of "~2 molecules" may also need to be revisited.

Author's Response: We thank the reviewer's careful reading and kind suggestions. We performed comparison of depth-associated sensitivity by using the method of Svensson et al. between Paired-seq and other platforms (Materials and methods). This method takes advantage of the big amount of data from 15 scRNA-seq platforms together to model the relationship between detection limit and sequencing depth of scRNA-seq platforms to make a fair evaluation when the data of each platform on each sequencing depth are insufficient. The basic hypothesis of this method is that the detection limit of each scRNA-seq platform decreases as the increasing of sequencing depth, and reaches a plateau at a constant sequencing depth. This method provided relatively stable and uniform metrics for the comparison of various scRNA-seq platforms, and achieved the detection limit of such as "10 molecules" for Drop-seq, "5 molecules" for inDrop, and also "2 molecules" for Paired-seq. However, this algorithm could not provide an absolute value of sensitivity but a relative value just for comparison, which brought confusion to reviewers. To remove this confusion, we have deleted this claim of 2 molecules detection limit and changed

the expression to “The sensitivity of Paired-seq is comparable to other modern single-cell approaches” in the revision.

2. Again, the needs of the single cell genomics field have matured beyond simple demonstrations on cell lines to processing fragile tissues where cell viability is limited and decreases with handling time. It is simply not enough to demonstrate the ‘paired-seq’ technique on cell lines where cells are hardy and well behaved. The authors claim that loading time of 15 min and 40 mins (indicated in referee response but not mentioned in manuscript) does not affect the number of detected genes in k562 and 3T3 cells. But this can be significantly detrimental to viability for cells harvested from primary tissues, that will be seen not directly from # genes detected but from higher expression of ribosomal and stress response genes. I am also not convinced that a ‘one size fits all’ chip design will work for cells from primary tissues where different cell types come in different cell-sizes.

Author’s Response: We thank the reviewer’s careful reading and kind suggestions. To further evaluate whether the shear force did more damage to the cells with the increase of loading time, we analyzed the expression levels of 9 genes (ARF1, CAST, CDK7, DBI, DDIT3, ENO2, ETF1, PLOD2 and RGS2) reported to have correlations with mechanical stress (J. Periodont Res. 2007, 42, 15–22). Herein, the 9 genes were biologically well characterized in terms of protein function, including cell communication, cell signaling, cell cycle, stress response and calcium release. From Supplementary Figure 9B, there were no remarkable differences between the samples with different loading time, indicating that the shear force did no damage to the cells.

Additionally, mES cells are tissue-derived cells that are taken from the mouse embryo tissue at 14.5 days of pregnancy with uneven cell size. The sequencing results of multiple parallel samples proved that there was no batch effect between different experimental samples, and the results of differentiation characterization from sequencing data were also consistent with the results by inDrop.

In order to further intuitively verify ‘one size fits all’, we measured the size of captured cells shown in Supplementary Figure 2A. The cell diameter was measured as follows: 24.5 μm, 18.6 μm, 13.2 μm, 14.8 μm, 23.4 μm from left to right for the upper cell capture channel and 8.6 μm, 15.0 μm, 16.4 μm, 5.5 μm, 20.8 μm from left to right for the lower cell capture channel. What is

more, taking into consideration of the previous suggestion about Nuc-seq, we used the Nuclei Extraction Kit (ExKine™ Nuclei Extraction Kit, KTP4002) to extract the nucleus, then stained it with HOECHST, mixed it with the SW480 cell line (stained with HOECHST for nuclei and DIL dye for cell membrane), and injected the mixture into the chip for single cells/ nuclei capture. The result showed that with the width of 3.58 μm gap, SW480 cell line (**Figure 1B**) and nucleus (**Figure 1C**) could be simultaneously captured by Paired-seq chip, demonstrating the ability to capture different sizes of cells/nuclei on a single chip and the feasibility to conduct Nuc-seq on the chip.

According to the results described above, we have fully demonstrated that we can capture and analyze cells of various sizes on a single chip, namely, ‘one size fits all’.

Figure 1. Characterization of chip size (A) and cell/ nucleus capture (B and C). Nuclei of SW480 cells were extracted by Nuclei Extraction Kit (ExKine™ Nuclei Extraction Kit, KTP4002). Cell nuclei were stained with HOECHST (blue), and cell membrane were stained with DIL dye (red).

3. The manuscript still does not have enough experimental details on how the chip is operated. What kind of pumps, valves and equipment were used for the gas phase flow? How was the chip incubated? What volumes of enzymes were needed to load the 800-cell and 2000-cell chip? What are the dead volumes? CAD files for the chip and schema for the equipment would be useful to understand the setup and workflow.

Author's Response: We thank the reviewer's careful reading and kind suggestions. We have described the operation of the chip in more detail in the attachment *Step by Step Protocol for*

Paired-seq and provided the CAD files of the chip design in the attachment. These files will be published as supporting materials.

4. *'Paired-seq' is a sophisticated technique. The manuscript is otherwise well-written and the authors have made substantial effort in comparing their technique to other single cell techniques. I am not convinced that the manuscript as it stands is impactful and of broad interest to the audience of Nature Communications, but it may be well suited for publication in Nature Biotechnology or Nature Genetics where practitioners of the single-cell genomics field may find the technique useful.*

Author's Response: We thank the reviewers' positive remarks on our work and recommendation to Nature Biotechnology or Nature Genetics. However, Paired-seq combined the microfluidic platforms with sequencing techniques to provide a significant tool for single-cell heterogeneity analysis, which involves areas of biotechnology, biochemistry, molecular biology and materials and thus is quite suitable for a multidisciplinary journal of *Nature Communications*.

Reviewers' Comments:

Reviewer #1:

Remarks to the Author:

The authors have adequately addressed my main concerns--most especially, revising the unjustified claims on capture sensitivity. They have also provided a step-by-step protocol and microfluidic chip design, which will aid the community in future technology development work. I see the paper as now supporting the stated claims with appropriate evidence

Reviewers' comments:

Reviewer #1 (Remarks to the Author):

1. *The authors have adequately addressed my main concerns--most especially, revising the unjustified claims on capture sensitivity. They have also provided a step-by-step protocol and microfluidic chip design, which will aid the community in future technology development work. I see the paper as now supporting the stated claims with appropriate evidence.*

Author's Response: We thank all efforts the reviewers spent on our manuscript, which greatly enhanced the quality of this work.